# Disease-associated mutations in Niemann-Pick type C1 alter ER calcium signaling and neuronal plasticity

Scott A. Tiscione[1], Oscar Vivas[1], Kenneth S. Ginsburg[2], Donald M. Bers[2], Daniel S. Ory[3], Luis F. Santana[1], Rose E. Dixon[1], and Eamonn J. Dickson[1]

Niemann-Pick type C1 (NPC1) protein is essential for the transport of externally derived cholesterol from lysosomes to other organelles. Deficiency of NPC1 underlies the progressive NPC1 neurodegenerative disorder. Currently, there are no curative therapies for this fatal disease. Given the $Ca^{2+}$ hypothesis of neurodegeneration, which posits that altered $Ca^{2+}$ dynamics contribute to neuropathology, we tested if disease mutations in NPC1 alter $Ca^{2+}$ signaling and neuronal plasticity. We determine that NPC1 inhibition or disease mutations potentiate store-operated $Ca^{2+}$ entry (SOCE) due to a presenilin 1 (PSEN1)–dependent reduction in ER $Ca^{2+}$ levels alongside elevated expression of the molecular SOCE components ORAI1 and STIM1. Associated with this dysfunctional $Ca^{2+}$ signaling is destabilization of neuronal dendritic spines. Knockdown of PSEN1 or inhibition of the SREBP pathway restores $Ca^{2+}$ homeostasis, corrects differential protein expression, reduces cholesterol accumulation, and rescues spine density. These findings highlight lysosomes as a crucial signaling platform responsible for tuning ER $Ca^{2+}$ signaling, SOCE, and synaptic architecture in health and disease.

## Introduction

Lysosomes are membrane-bound acidic organelles found in every eukaryotic cell. Historically known as catabolic sites for degradation and recycling of waste products, our understanding of their functional responsibilities has dramatically expanded and includes essential roles for nutrient sensing, transcriptional regulation, metabolic homeostasis, and $Ca^{2+}$ signaling (Xu and Ren, 2015; Saxton and Sabatini, 2017). Their dynamic positioning and growing repertoire of cellular functions have elevated the lysosome from "garbage can" to a signaling organelle exquisitely positioned to influence cellular growth and survival. Accumulating evidence suggests that dysfunction of endolysosomal and autophagic pathways is associated with the progression of neurodegenerative diseases, such as Alzheimer's, Parkinson's, and inherited lysosomal storage disorders characterized by the intralysosomal buildup of partially degraded metabolites (Laplante and Sabatini, 2012; Fraldi et al., 2016; Castellano et al., 2017). Despite the involvement of lysosomes in neurodegenerative disease progression, the molecular mechanisms that link lysosomes to neuronal dysfunction have not been fully established.

A common metabolite that excessively accumulates across several neurodegenerative disorders is the organic sterol cholesterol (Liu et al., 2010). A significant portion of cellular cholesterol is exogenously imported in lipoproteins via clathrin-mediated endocytosis. As internalized vesicles mature into late-endosomes/lysosomes, acid lipases free cholesterol from lipoproteins. Consequently, luminal Niemann-Pick type C2 (NPC2) proteins deliver cholesterol to membrane-spanning NPC1 proteins to facilitate cholesterol transport across the late-endosome/lysosome membrane. Following its transport to the cytoplasmic leaflet of late-endosome/lysosome membranes, sterol transfer proteins such as ORP5 and ORPL1 act to transfer cholesterol to the ER (Du et al., 2011; Zhao and Ridgway, 2017). Underscoring the importance of NPC1-mediated cholesterol egress is the fatal neurodegenerative disorder NPC1 disease. NPC1 disease arises from a loss-of-function mutation in the NPC1 protein that results in a massive accumulation of cholesterol in lysosomes and significant perturbation of cholesterol distribution and homeostasis (Millard et al., 2000; Vanier and Millat, 2003). Despite the correlation between dysfunctional cholesterol homeostasis and neuronal cell death in NPC1 disease, the molecular mechanisms that facilitate neurodegeneration are currently unknown.

The $Ca^{2+}$ hypothesis of neurodegeneration postulates that sustained disturbances in $Ca^{2+}$ signaling are pivotal for the pathogenesis of various neurodegenerative disorders (Berridge, 2010). Indeed, neuronal $Ca^{2+}$ signaling plays a fundamental role in a wide variety of events, including electrical excitability,

.................................................................................................................................................................................................................................
[1]Department of Physiology and Membrane Biology, University of California, Davis, Davis, CA; [2]Department of Pharmacology, University of California, Davis, Davis, CA; [3]Department of Internal Medicine, Washington University School of Medicine, St. Louis, MO.

Correspondence to Eamonn J. Dickson: ejdickson@ucdavis.edu.



synaptic plasticity, gene transcription, and survival (Berridge, 2010; Brini et al., 2014; Wu et al., 2016; Pchitskaya et al., 2018). In neurons, the main intracellular source of $Ca^{2+}$ comes from the ER. Triggered $Ca^{2+}$ release from the ER can be accomplished by hormone or neurotransmitter binding to $G\alpha_q$-coupled receptors on the plasma membrane (PM). Following the release of ER $Ca^{2+}$, two major mechanisms serve to refill depleted stores: (1) sarco-ER calcium ATPases (SERCA) and (2) store-operated $Ca^{2+}$ entry (SOCE). Classically, the molecular identity of the proteins choreographing SOCE are STIM proteins in the ER membrane and ORAI channels in the PM (Prakriya and Lewis, 2015). Following ER $Ca^{2+}$ depletion, STIM proteins sense the reduction in ER luminal $Ca^{2+}$, subsequently oligomerize, and translocate to ER–PM contact sites, where they activate $Ca^{2+}$-conducting ORAI channels; this results in the flux of extracellular $Ca^{2+}$ into the cell, where it is then sequestered into the ER via SERCA. Accumulating evidence suggests that dysregulation of SOCE in neurons perturbs intracellular $Ca^{2+}$ signaling and contributes to the pathogenesis of neurodegenerative disorders (Sun et al., 2014; Wu et al., 2016; Bollimuntha et al., 2017; Secondo et al., 2018). Given reports that ER $Ca^{2+}$ pathways are dysregulated across several neurodegenerative diseases and the correlation between altered cholesterol homeostasis and neurodegeneration, we hypothesized that the aberrant cholesterol efflux in NPC1 disease may perturb ER $Ca^{2+}$ signaling pathways.

Here, we describe a molecular pathway linking cholesterol egress from lysosomes to the tuning of ER $Ca^{2+}$ and SOCE, which influences cytoplasmic $Ca^{2+}$ concentrations and synaptic plasticity in neurons.

## Results

### Cholesterol and $Ca^{2+}$ are misregulated in NPC1$^{I1061T}$ disease

The most common NPC disease mutation, NPC1$^{I1061T}$, results in misfolding of the NPC1 protein, its subsequent targeting for degradation (Gelsthorpe et al., 2008), and thus its functional loss. Blotting for NPC1 revealed the anticipated 80% reduction in NPC1 protein from NPC1$^{I1061T}$ patient fibroblasts compared with healthy patient fibroblasts (Fig. 1 A). The primary characteristic of NPC disease is the massive accumulation of cholesterol in lysosomes. To confirm this cellular phenotype, we stained for unesterified cholesterol using the naturally fluorescent filipin compound. Fig. 1 B shows representative superresolution images of filipin-stained healthy and NPC1$^{I1061T}$ fibroblasts; qualitatively, the black vesicular structures represent cholesterol unable to exit lysosomes. These data show that the lysosomal NPC1 protein and, as a consequence, lysosomal cholesterol homeostasis are significantly perturbed in NPC1$^{I1061T}$ disease fibroblasts.

Next, we tested the hypothesis that $Ca^{2+}$ dynamics are misregulated in NPC1$^{I1061T}$ disease fibroblasts. To this end, healthy and NPC1$^{I1061T}$ fibroblasts were loaded with cytosolic $Ca^{2+}$ indicator Fluo-4 and subjected to a protocol with the SERCA inhibitor thapsigargin (TG) in an external 0 mM $Ca^{2+}$ solution, followed by a 2 mM $Ca^{2+}$-containing solution. This experimental protocol allows us to analyze the $Ca^{2+}$ released from the ER (as the area under the curve [AUC] between 150 s and 450 s) and SOCE (AUC >450 s). We found that NPC1$^{I1061T}$ fibroblasts have

significantly altered $Ca^{2+}$ dynamics (Fig. 1 C). Quantitative analyses determined NPC1$^{I1061T}$ fibroblasts have enhanced $Ca^{2+}$ influx, less $Ca^{2+}$ release from the ER, and larger, more rapid SOCE following TG treatment (Fig. 1, D and E). The consequences of these alterations in $Ca^{2+}$ dynamics are elevated resting cytoplasmic $Ca^{2+}$ levels, as indicated by a significant increase in Fluo-4 intensity (Fig. 1 F) and Fura-2 ratios (Fig. 1 G). Calibration of these cytosolic $Ca^{2+}$ dyes revealed that NPC1$^{I1061T}$ fibroblasts have significantly increased cytoplasmic $Ca^{2+}$ concentrations (healthy, 153 nM ± 10 nM; NPC1$^{I1061T}$, 230 nM ± 7 nM; $n = 6$). To fully underscore that these alterations in $Ca^{2+}$ signaling were due to NPC1 dysfunction and not to adaptation of this patient cell, we measured $Ca^{2+}$ dynamics under four additional conditions of NPC1 dysfunction: two additional NPC1 disease patient cell lines (NPC1$^{I1061T/I1061T}$ and NPC1$^{I1061T/P1007A}$), NPC1$^{-/-}$ cells, and mouse embryonic fibroblasts (MEFs) treated overnight with an inhibitor of NPC1 (U18666A [UA]; 1 μM; Lu et al., 2015). Across each experimental condition, TG-mediated $Ca^{2+}$ release was reduced and SOCE was elevated (Fig. S1). Collectively, these data show lysosomal cholesterol is associated with altered $Ca^{2+}$ dynamics in NPC1 disease and support the hypothesis that NPC1-mediated cholesterol efflux tunes $Ca^{2+}$ homeostasis.

### NPC1$^{I1061T}$ fibroblasts have greater endogenous ORAI-STIM $Ca^{2+}$ influx

To determine the molecular identity of the enhanced $Ca^{2+}$ influx into NPC1$^{I1061T}$ cells, we transfected cells with a genetically encoded $Ca^{2+}$ sensor targeted to the PM (GCaMP-CAAX; Fig. 2, A and B) and studied the effects of pharmacological blockade of candidate channels. Fig. 2, A and B, shows representative confocal images of healthy and NPC1$^{I1061T}$ patient fibroblasts transfected with GCaMP-CAAX, focused at the PM footprint. Measurement of the resting PM intensity revealed a 2.6-fold (±0.7) higher intensity of NPC1$^{I1061T}$ fibroblasts, suggesting greater $Ca^{2+}$ influx across the PM. To test if STIM–ORAI interactions represent the molecular identity of this differential $Ca^{2+}$ flux, we treated cells with an inhibitor of ORAI1 channels, AnCoA4 (Sadaghiani et al., 2014). Measurement of GCaMP-CAAX fluorescent intensities at the PM revealed a significantly greater AnCoA4-sensitive component from NPC1$^{I1061T}$ fibroblasts (healthy, 49% reduction ±6.7%; NPC1$^{I1061T}$, 75% reduction ±3.5%; Fig. 2 C). Patch-clamp electrophysiology recordings confirmed a larger AnCoA4-sensitive calcium release–activated channel current ($I_{CRAC}$) conductance from NPC1$^{I1061T}$ fibroblasts (Fig. S2 A) and suggests enhanced resting ORAI1 conductance in NPC1 disease cells.

To determine if enhanced abundance or altered ORAI1 and STIM1 distribution contributes to increased ORAI1-dependent $Ca^{2+}$ conductance in NPC1$^{I1061T}$ disease we performed Western blot analysis and superresolution nanoscopy. Western blot analysis revealed significant increases in total protein levels of ORAI1 (33% ± 11%) and STIM1 (20% ± 4%) from NPC1$^{I1061T}$ fibroblasts relative to healthy patient cells (Fig. 2, D and E), while STIM2 protein expression was not significantly altered (Fig. S2 C). Next, using superresolution total internal reflection fluorescence (TIRF) microscopy, we generated localization maps of

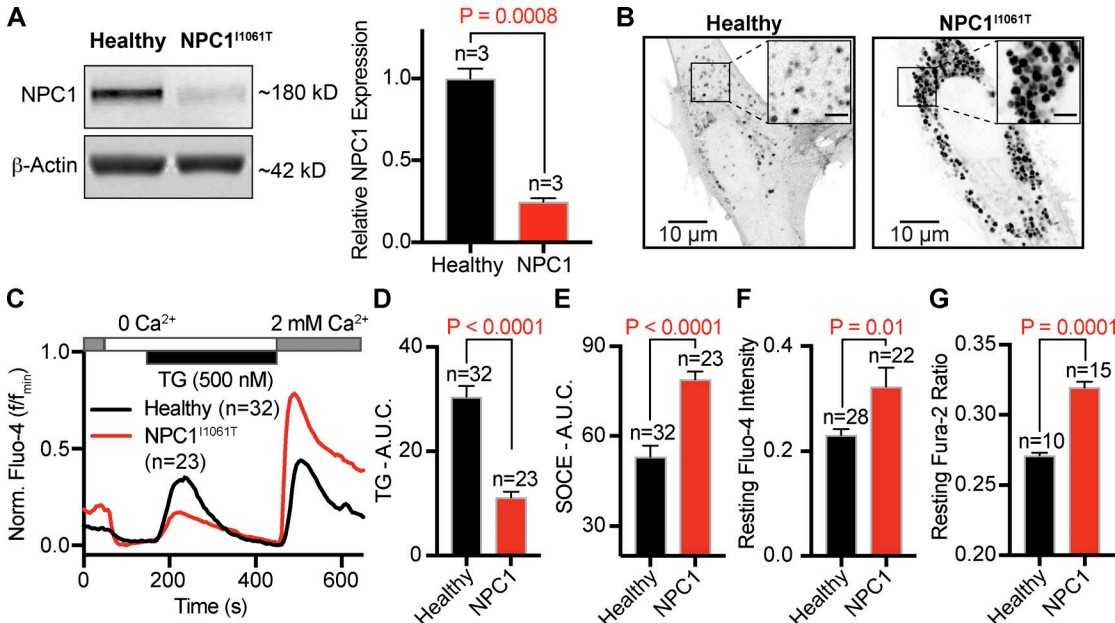

Figure 1. **NPC1 disease fibroblasts have altered cholesterol homeostasis and Ca²⁺ signaling. (A)** Left: Representative Western blot of NPC1 in healthy and NPC1^I1061T disease fibroblasts. Right: Quantification of NPC1 protein expression, normalized to β-actin. **(B)** Representative superresolution images of healthy and NPC1^I1061T fibroblasts fixed and stained with filipin to show cholesterol distribution. Scale bar of inset represents 2.5 μm. **(C)** Representative time series from Fluo-4–loaded healthy and NPC1^I1061T fibroblasts. **(D–F)** Quantification of Fluo-4 signals from healthy and NPC1^I1061T fibroblasts. **(G)** Comparison of resting fura-2 ratios. Error bars represent the standard error of the mean.

endogenous ORAI1 and STIM1 proteins to determine if increased protein levels correlated to altered size and distribution of puncta. Fig. 2 F shows representative TIRF and superresolution TIRF images of immunolabeled ORAI1 in healthy and NPC1^I1061T fibroblasts. Quantitative analyses of superresolution localization maps revealed the abundance and average size of ORAI1 puncta are increased (Figs. 2 G and S2 D) in NPC1^I1061T fibroblasts. For STIM1 (Fig. 2 H), we also found a similar trend of increased abundance (Fig. 2 I) and average size (Figs. 2 I and S2 D) of puncta from NPC1^I1061T TIRF localization maps. Collectively, these data support the interpretation that Ca²⁺ influx via ORAI1 is higher in NPC1^I1061T patient cells, with multiple assays providing evidence that altered STIM1/ORAI1 protein abundance and distribution contribute to this increase.

## NPC1^I10161T patient cells have increased size and number of resting STIM1 and ORAI1 puncta

Given the enhanced Ca²⁺ influx (Fig. 2, A–C; and Fig. S2 A) and increased STIM1 and ORAI protein levels/distribution (Fig. 2, D–I) in NPC1^I1061T fibroblasts, we wanted to determine if the number of interactions between STIM1 and ORAI1 puncta at rest were also increased. To this end, we used superresolution microscopy to measure the degree of colocalization between endogenous STIM1 and ORAI1 in healthy and NPC1 patient fibroblasts. Superresolution images revealed that the degree of overlapping pixels between ORAI1 and STIM1 was significantly elevated in NPC1^I1061T fibroblasts relative to healthy cells (compare Fig. 3, A and B). Quantitative analysis of superresolution images revealed an approximately threefold increase in the percentage of overlapping pixels, with both STIM1/ORAI1

puncta area and density significantly increased (Fig. 3 C). Next, we investigated whether STIM1/ORAI1 puncta formation occurred more rapidly in NPC1^I1061T fibroblasts. To this end, we expressed mCherry-STIM1 and Orai1-EGFP and monitored their dynamics before and during TG application. Fig. S2 F shows representative images of resting healthy and NPC1^I1061T fibroblasts transfected with ORAI1-GFP and mCherry-STIM1. As expected, NPC1^I1061T fibroblasts displayed larger and more abundant STIM1/ORAI1 puncta in resting cells. Depletion of ER Ca²⁺ stores with a 5-min application of TG revealed an acceleration in STIM1/ORAI1 puncta formation in NPC1^I1061T fibroblasts (Fig. S2, G and H), which also appeared to be larger than their healthy cell counterparts (Fig. S2 I). These data suggest that STIM1 and ORAI1 are not only pre-associated in resting NPC1^I1061T fibroblasts but also may form puncta more quickly following depletion of intracellular Ca²⁺ stores.

## ER Ca²⁺ is reduced in NPC1^I1061T fibroblasts

Having established that SOCE and STIM1/ORAI1 clustering are enhanced in NPC1^I1061T fibroblasts, we next tested if this occurs as a consequence of reduced ER luminal Ca²⁺ levels (Fig. 4 A). A reduction in ER Ca²⁺ levels would decrease the free Ca²⁺ concentration available to bind STIM1's EF-hand domain and consequently increase the likelihood of STIM1 activation. To measure ER Ca²⁺, healthy and NPC1^I1061T patient fibroblasts were transfected with D1-ER (Palmer et al., 2004), a genetically encoded Ca²⁺ sensor targeted to the lumen of the ER. Ca²⁺ binding to the probe increases Förster resonance energy transfer (FRET) between CFP and YFP fluorophores to increase the FRET ratio (Fig. 4 B). Fig. 4 C shows representative images of FRET ratios in

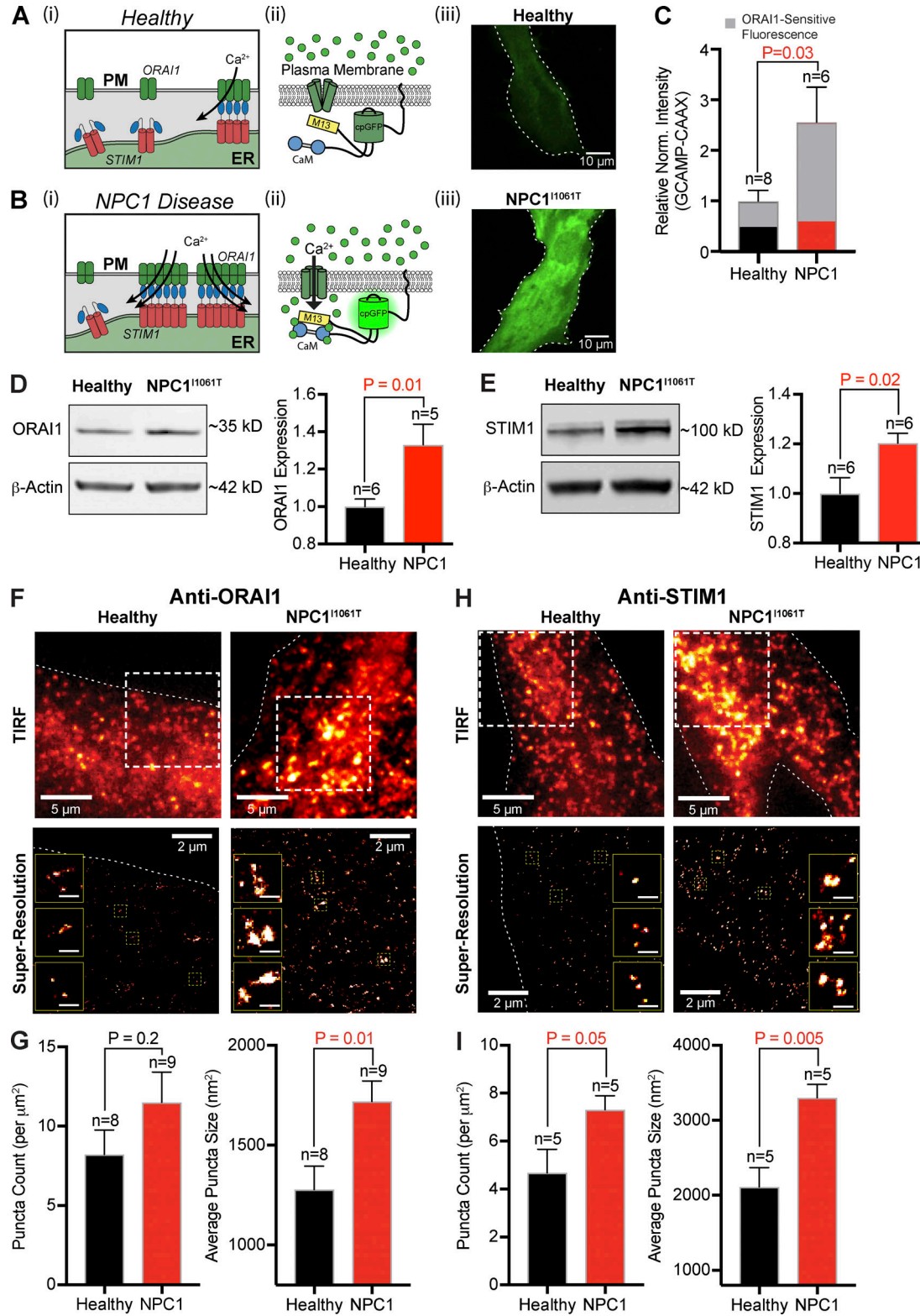

Figure 2. **Resting SOCE, coupled with STIM1/ORAI1 expression and distribution, is increased in NPC1^{I1061T} fibroblasts. (A)** Diagram of SOCE in healthy cells (i), schematic of GCAMP-CAAX Ca^{2+} probe in healthy cells (ii), and representative live confocal images of a healthy cell expressing GCAMP-CAAX (iii). **(B)** Same as A, only for NPC1^{I1061T} fibroblasts. **(C)** Quantification of resting GCAMP-CAAX fluorescence. Gray portion indicates the AnCoA4-sensitive reduction in resting fluorescence. **(D)** Left: Representative ORAI1 Western blot from healthy and NPC1^{I1061T} fibroblasts. Right: Quantification of ORAI1 protein expression in fibroblasts, normalized to β-actin. **(E)** Same as D, only for STIM1. **(F)** Top: Representative TIRF images from healthy and NPC1^{I1061T} fibroblasts stained for anti-ORAI1. Bottom: Superresolution localization map from dashed boxes within the TIRF images; dashed lines delineate edges of cells. Inset: Representative ORAI1 puncta. Scale bar represents 0.25 μm. **(G)** Quantification of anti-ORAI1 localization map. **(H and I)** Same as F and G, only for STIM1. Error bars represent the standard error of the mean.

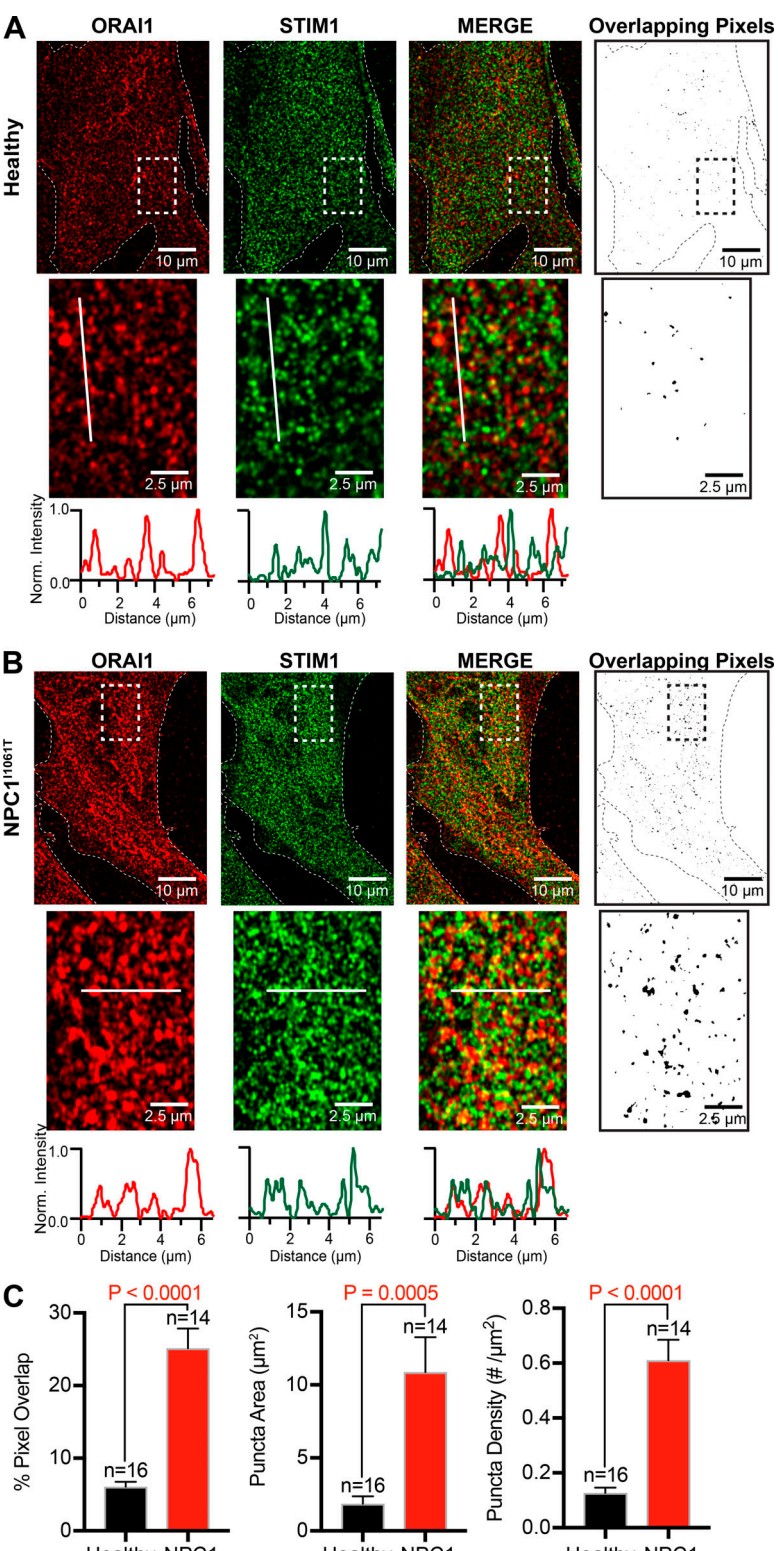

Figure 3. **Size and number of STIM1/ORAI1 puncta are enhanced in NPC1[I1061T] fibroblasts. (A)** Left: Representative superresolution image of a healthy fibroblast immunostained for ORAI1 (red), STIM1 (green), and merged (yellow). Right: Binary colocalization map of overlapping pixels from ORAI1 and STIM1 channels. Bottom: Zoomed regions denoted by dashed boxes and representative line plots taken from each solid white line. **(B)** Same as A, only for an NPC1[I1061T] fibroblast. **(C)** Quantification of superresolution images. Error bars represent the standard error of the mean.

resting healthy and NPC1[I1061T] fibroblasts transfected with this probe. Quantification of cells expressing the probe revealed a significant reduction in FRET ratio of ∼25% in NPC1[I1061T] fibroblasts (Fig. 4 D), suggesting reduced ER Ca²⁺ levels. Calibration of the D1-ER FRET probe revealed healthy cells as having a luminal ER free Ca²⁺ concentration of ∼270 µM (healthy,

268 µM ± 15 µM; $n = 5$), while NPC1[I1061T] cells at a significantly lower ER Ca²⁺ concentration of ∼170 µM (NPC1[I1061T] 167 µM ± 26 µM; $n = 6$). To confirm this observation, we measured the total Ca²⁺ released from intracellular stores following application of the Ca²⁺ ionophore ionomycin (in the absence of extracellular Ca²⁺). Experiments from Fluo-4–loaded cells treated

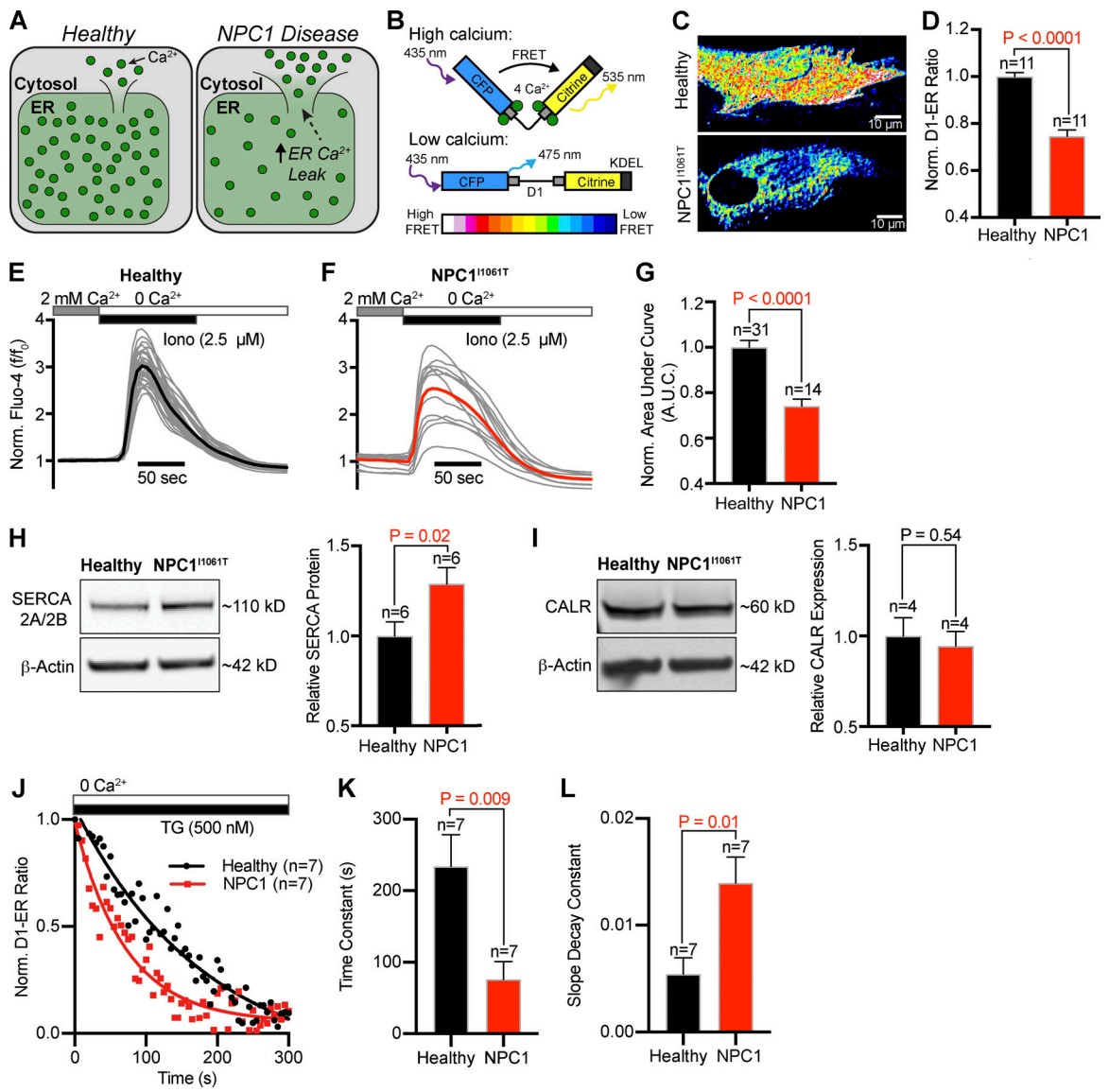

Figure 4. **Resting luminal ER Ca²⁺ is reduced in NPC1^{I1061T} fibroblasts. (A)** Graphical depiction of differential ER Ca²⁺, mediated by enhanced ER Ca²⁺ leak in NPC1^{I1061T} fibroblasts. **(B)** Schematic of D1-ER Ca²⁺ probe. **(C)** Representative live confocal images of healthy and NPC1^{I1061T} fibroblasts expressing D1-ER. **(D)** Quantification of D1-ER FRET ratios. **(E)** Normalized individual Fluo-4 intensities (gray) and averaged intensity (black, n = 31) from healthy fibroblasts following addition of a Ca²⁺-free ionomycin-containing external solution. **(F)** Same as E, only NPC1^{I1061T} fibroblasts (red, n = 14). **(G)** Quantification of the normalized area under the ionomycin curve. **(H)** Left: Representative Western blot for SERCA 2A/2B from healthy and NPC1^{I1061T} lysates. Right: Quantification of protein expression, normalized to β-actin. **(I)** Same as H, only for calreticulin (CALR). **(J)** Normalized D1-ER FRET ratios from healthy (black circles) and NPC1^{I1061T} fibroblasts (red squares) following application of TG. Fitted lines (healthy: black; NPC1^{I1061T}: red) represent one-phase decay curves. **(K and L)** Quantitative comparison of the slope decay and time constant from the healthy and NPC1^{I1061T} nonlinear regression curves. Error bars represent the standard error of the mean.

with ionomycin (2.5 µM) revealed a significant reduction in the pool of Ca²⁺ mobilized to the cytoplasm in NPC1^{I1061T} fibroblasts (Fig. 4, E–G). Together, these experiments strongly suggest ER luminal Ca²⁺ concentrations are reduced in NPC disease.

Next, we wanted to determine if the reduction of ER luminal Ca²⁺ from NPC1^{I1061T} fibroblasts was due to transcriptional changes in protein expression. ER luminal Ca²⁺ concentration is controlled by three main parameters: (1) Ca²⁺ buffering proteins, (2) Ca²⁺ reuptake pumps, and (3) rate of intrinsic Ca²⁺ leak. We hypothesized that loss of NPC1 function may alter the protein levels underlying one or more of these parameters. To test this

hypothesis, we performed Western blots for several proteins that may alter ER Ca²⁺ load. We began with SERCA, the ER's primary pathway of refilling and maintenance of luminal Ca²⁺. Western analysis of SERCA isoforms 2A/2B revealed a significant increase of 29% ± 9% in NPC^{II1061T} fibroblasts (Fig. 4 H), making it unlikely to be directly involved in regulating the enhanced SOCE in NPC disease. Next, we tested Calreticulin, an important luminal ER Ca²⁺ buffering protein, and found it not significantly altered at the protein level in NPC1^{I1061T} fibroblasts (Fig. 4 I). Finally, we tested the hypothesis that NPC^{II1061T} fibroblasts have a leakier ER compared with healthy patient cells

(see hypothesis; Fig. 4 A). To measure the rate of intrinsic Ca²⁺ leak from the ER, healthy and NPC1$^{I1061T}$ fibroblasts were transfected with the D1-ER FRET probe and treated with the SERCA pump inhibitor TG (500 nM) in the absence of extracellular Ca²⁺. Decreases in FRET ratio following inhibition of the SERCA pump represent Ca²⁺ leaking from the ER (Fig. 4 J). Fitting healthy and NPC1 FRET ratios with one-phase decay regression curves revealed that NPC1$^{I1061T}$ fibroblasts have a significantly faster time constant of decay (Fig. 4, K and L). Collectively, these data suggest that NPC1$^{I1061T}$ fibroblasts have a greater intrinsic ER Ca²⁺ leak, explaining why their ER free Ca²⁺ concentration is significantly lower than healthy cells.

## Presenilin 1 (PSEN1) mediates the decrease in ER Ca²⁺ and enhanced SOCE in NPC1 disease

PSEN1 is one of the core proteins of the γ-secretase complex that has been suggested to potentiate ER Ca²⁺ channels (Leissring et al., 1999) and represent the molecular identity of the low-conductance ER-leak channels (Nelson et al., 2011; Supnet and Bezprozvanny, 2011). PSEN1 undergoes endoproteolytic processing to produce two fragments, an ∼30-kD N-terminal fragment and ∼20-kD C-terminal fragment. Western blot analysis revealed that both fragments are significantly elevated in NPC1$^{I1061T}$ fibroblasts (Fig. 5, A and B). Despite this elevation in fragments, it is reported that the full-length (∼50 kD) PSEN1 holoprotein functions as the Ca²⁺ leak channel (Tu et al., 2006). Due to its low abundance and short half-life (Ratovitski et al., 1997), we, like others (Walter et al., 1996; Citron et al., 1997; Ratovitski et al., 1997), failed to detect a holoprotein band in patient fibroblasts. To overcome this limitation, we overexpressed the full-length PSEN1 protein and found that inhibiting NPC1 cholesterol efflux overnight, with UA, significantly increased PSEN1 holoprotein levels (Fig. 5 C).

An increase in PSEN1 expression and/or activity is predicted to result in a leakier ER, which may account for the observed decrease in resting ER Ca²⁺ levels in NPC1 disease and contribute to enhanced STIM/ORAI aggregation and SOCE. To test whether up-regulation of PSEN1 is responsible for these altered Ca²⁺ phenotypes in NPC disease, we knocked down PSEN1 in NPC1$^{I1061T}$ fibroblasts and measured ER Ca²⁺ levels and SOCE (Fig. 5, D–H). 48-h transfection with siRNA directed against PSEN1 significantly increased the ionomycin- (Fig. 5, D and E) and TG-mediated (Fig. 5, F and G) elevations in Ca²⁺ from NPC fibroblasts compared with NPC fibroblasts transfected with a negative nontargeting control. Additionally, the SOCE AUC was significantly reduced in the PSEN1 knockdown fibroblasts compared with controls (Fig. 5, F and H). Further evidence of PSEN1's ability to influence ER Ca²⁺ levels is found in Fig. S3, where we show that treating healthy cells with PSEN1 siRNA increases ER Ca²⁺ (Fig. S3, A–C), overexpression of the catalytically inactive PSEN1$^{D257A}$ (accumulates holoprotein form) decreases ER Ca²⁺ levels and increases the rate of Ca²⁺ leak from the ER (Fig. S3, D–F), and PSEN$^{-/-}$ cells have increased TG-mediated Ca²⁺ release (Fig. S3, G–I). These data align with PSEN1's putative role as an ER Ca²⁺ leak channel. As a final piece of evidence that PSEN1 mediates the reduction in ER Ca²⁺ found in NPC1 disease, treating PSEN$^{-/-}$ cells with UA did not alter Ca²⁺

dynamics (Fig. S3, J–O). These findings suggest that changes in the expression of PSEN1 influence ER Ca²⁺ concentrations and contribute to reduced ER Ca²⁺ levels in NPC1 disease.

Pharmacological elevations in ER Ca²⁺ have been shown to adapt NPC1 proteostasis, increase the functional amount of NPC1 in lysosomal membranes, and rescue the aberrant storage of cholesterol (Yu et al., 2012). With this in mind, we tested the hypothesis that treating NPC1$^{I1061T}$ fibroblasts with PSEN1 siRNA to increase ER Ca²⁺ levels (Fig. 5, D–G) may increase the amount of NPC1 protein and alter cholesterol storage. Western blot analysis revealed that 24-h PSEN1 siRNA treatment of NPC1$^{I1061T}$ cells increased NPC1 protein approximately threefold (Fig. 5 I, J) and caused a corresponding decrease in the area and intensity of filipin-positive vesicles (Fig. 5, K–N). These results demonstrate that increases in PSEN1, which decrease ER Ca²⁺ levels, may compound cholesterol accumulation in NPC1 disease.

## Inhibition of SREBP rescues the Ca²⁺ and transcriptional changes of NPC1$^{I1061T}$ fibroblasts

Having established that PSEN1 may be part of the mechanism linking lysosomal cholesterol dysfunction to alterations in Ca²⁺ homeostasis in NPC disease, we next tested the hypothesis that activation of SREBP (sterol-response element–binding protein) is responsible for the enhanced expression of key Ca²⁺ handling proteins, including PSEN1, in NPC1 disease (Fig. 6 A). SREBPs are transcription factors that govern cellular lipogenesis and cholesterol metabolism. Inactive SREBP (p-SREBP) is associated with SREBP cleavage-activating protein (SCAP) and an INSIG family protein, which retains SREBP in the ER membrane in a cholesterol-dependent manner. When cholesterol content at the ER falls below a certain threshold (∼5 mol%; Infante and Radhakrishnan, 2017), such as in NPC1 disease, the SREBP–SCAP complex dissociates from INSIG and translocates to the Golgi, where S1P and S2P proteases cleave the SREBP precursor and release the mature, active form of SREBP (n-SREBP), which subsequently translocates to the nucleus to up-regulate gene transcription. Western blot analysis revealed that n-SREBP (active form) was twofold higher in NPC1$^{I1061T}$ fibroblasts relative to healthy patient fibroblasts (Fig. 6, B and C). To determine if SREBP impacts the transcription and translation of Ca²⁺ handling proteins, we incubated NPC1$^{I1061T}$ fibroblasts with 500 nM PF-429242, an inhibitor of S1P that prevents activation of SREBP (Fig. 6, B and C), for 24 h alongside vehicle controls and measured mRNA and protein levels, respectively. qPCR analysis of mRNA levels revealed PSEN1, STIM1, and ORAI1 are all elevated in NPC1 fibroblasts, with SREBP inhibition decreasing levels back to healthy levels (Fig. S4, A and B). Analysis of Western blots revealed that inhibition of SREBP also significantly reduced protein levels of SERCA, STIM1, ORAI1, and PSEN1 (Fig. 6, D and E). These results support the hypothesis that the SREBP pathway mediates transcriptional changes in Ca²⁺ signaling proteins in NPC1 disease.

Given the involvement of the SREBP pathway in influencing the expression of key Ca²⁺ signaling proteins in NPC1 disease, we tested if inhibition of SREBP and subsequent rescue of Ca²⁺ signaling protein levels restored Ca²⁺ dynamics. Consistent with

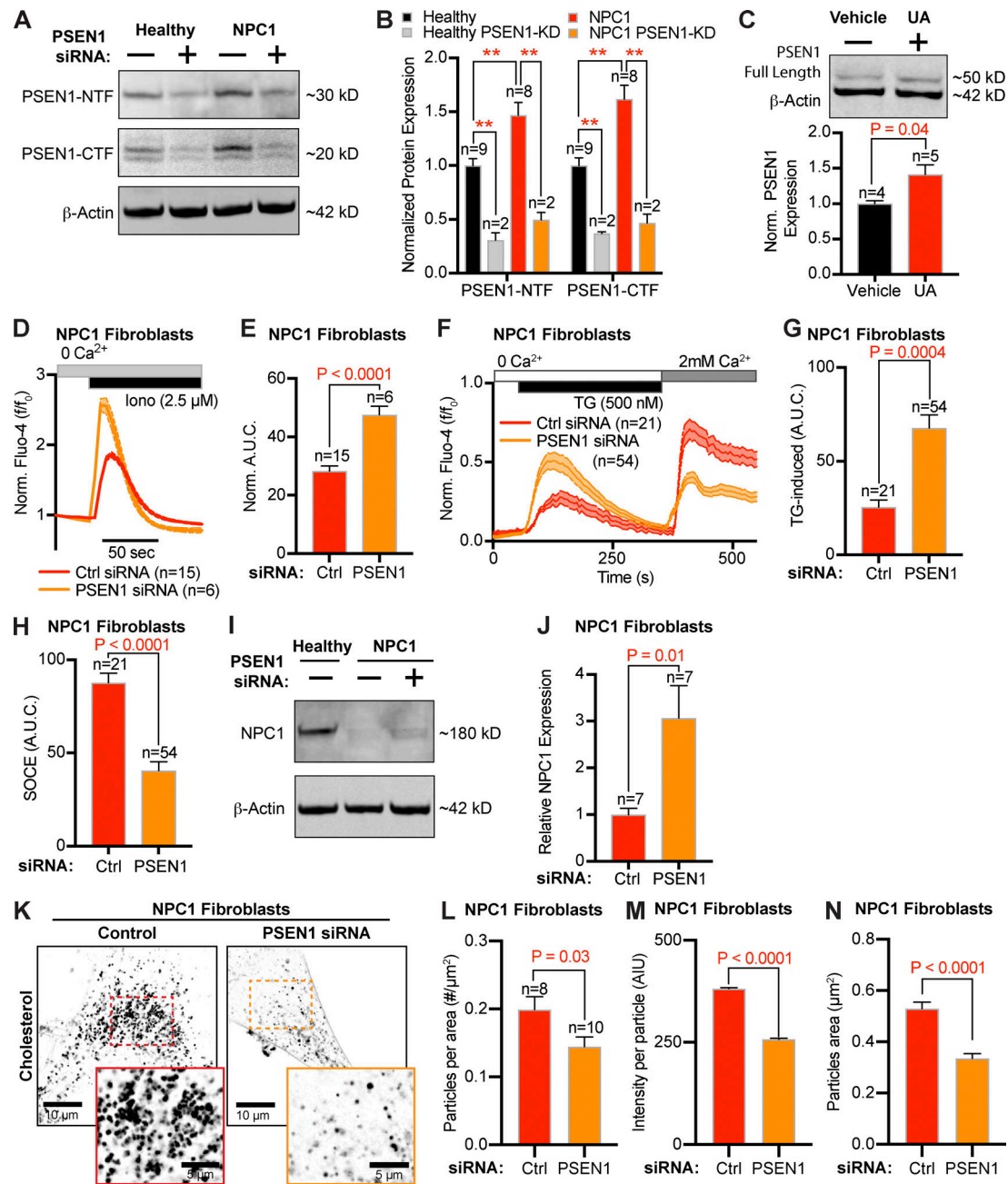

**Figure 5. Differential expression of PSEN1 alters ER Ca²⁺ and SOCE in NPC1^I1061T fibroblasts.** **(A)** Representative PSEN1-NTF and PSEN1-CTF Western blots from healthy and NPC1^I1061T cells with or without siRNA targeting PSEN1. **(B)** Quantification of differential protein expression, normalized to β-actin. **(C)** Top: Representative PSEN1 Western blot from tsA-201 cells transfected with PSEN1 and treated with NPC1 inhibitor, UA, or vehicle control. Bottom: Quantification of differential PSEN1 protein expression, normalized to β-actin. **(D)** Averaged intracellular Fluo-4 responses following ionomycin treatment in NPC1^I1061T fibroblasts transfected for 48 h with either negative control siRNA (red) or siRNA targeting PSEN1 (orange). **(E)** Quantification of AUC, from NPC1^I1061T fibroblasts transfected with PSEN1 (orange) or control (red) siRNA. **(F)** Averaged time-course of NPC1^I1061T fibroblasts transfected with control siRNA (red) or siRNA targeting PSEN1 (orange). **(G)** Quantification of the AUC during TG. **(H)** Quantification of the AUC during SOCE. **(I)** Representative Western blot of NPC1 in healthy and NPC1^I1061T fibroblasts with or without siRNA (24 h) targeting PSEN1. **(J)** Quantification of differential protein expression, normalized to β-actin. **(K)** Representative superresolution images of filipin staining from NPC1^I1061T fibroblasts treated with control or PSEN1 siRNA for 72 h. **(L–N)** Quantification of superresolution filipin images. Error bars represent the standard error of the mean.

this hypothesis, inhibition of NPC1 in SCAP⁻/⁻ cells, which lacks the ability to activate the SREBP pathway (Rawson et al., 1999), failed to alter Ca²⁺ dynamics (Fig. S4, C–E), while treatment of NPC1^I1061T fibroblasts with PF-429242 significantly increased the ionomycin-mediated elevation in cytoplasmic Ca²⁺

(Fig. 6, F and G), the TG-sensitive AUC (Fig. 6 H and I), the SOCE AUC (Fig. 6 J), and the SOCE/TG ratio (Fig. 6 K) back to healthy control levels. Appropriate control experiments detailing the effect of PF-429242 on healthy fibroblasts are found in Fig. S5, A–I.

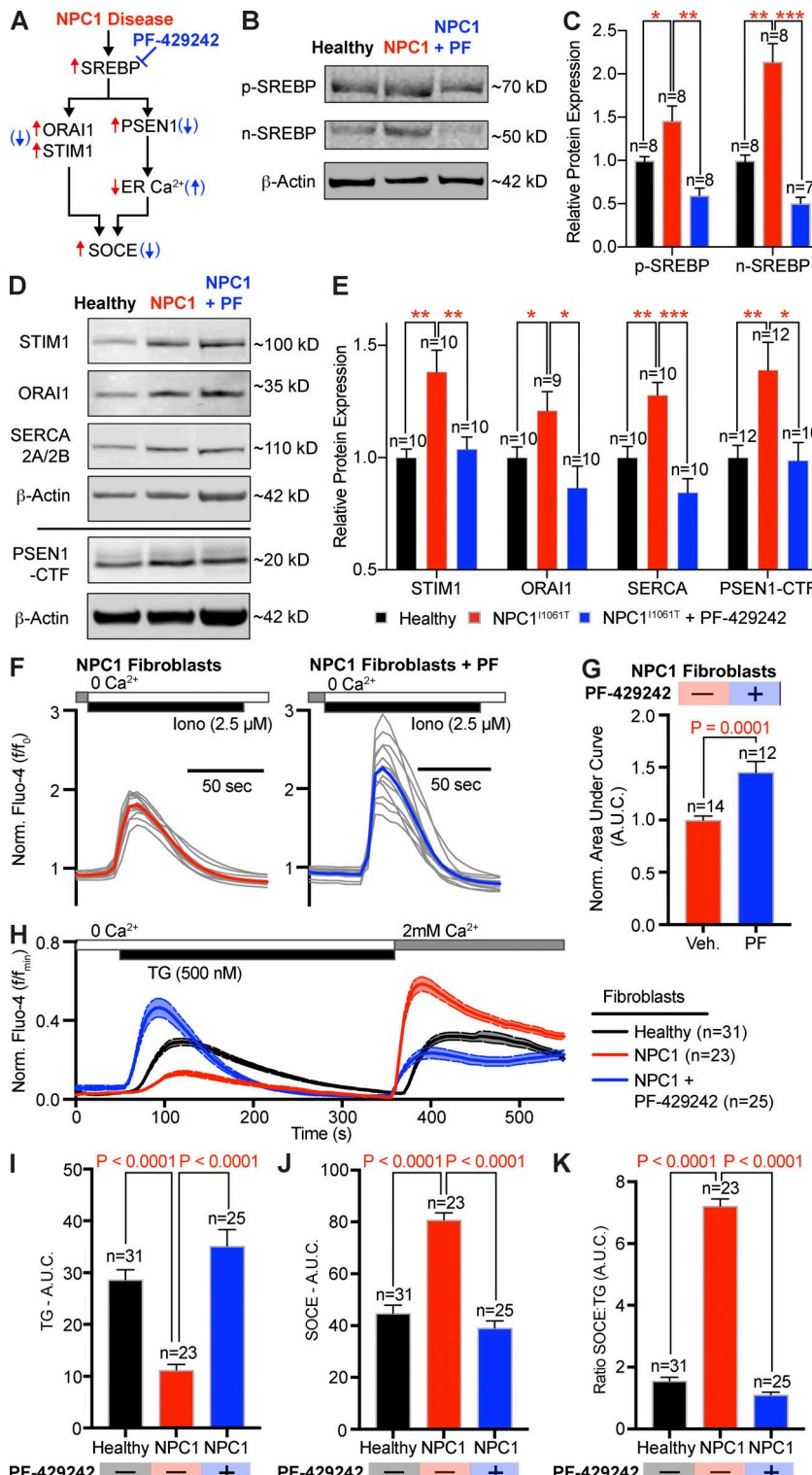

Figure 6. **SREBP-mediated control of PSEN1 and Ca²⁺ proteins tunes ER Ca²⁺ and SOCE in NPC1^I1061T fibroblasts.** **(A)** Proposed pathway for SREBP-mediated alterations of Ca²⁺ in NPC1 disease. Red arrows represent proteins/Ca²⁺ pathways increasing in NPC1 disease. Blue arrows represent changes following treatment with PF-429242. **(B)** Representative inactive SREBP (p-SREBP) and active SREBP (n-SREBP) Western blots from healthy fibroblasts, NPC1^I1061T fibroblasts, and NPC1^I1061T fibroblasts treated with SREBP inhibitor, PF-429242. **(C)** Quantification of differential protein expression, normalized to β-actin (*, $P < 0.5$; **, $P < 0.1$; ***, $P < 0.0001$). **(D)** Left: Representative Western blots showing SERCA, STIM1, ORAI1, and PSEN1-CTF protein in healthy fibroblasts and NPC1^I1061T fibroblasts, with or without 24-h pretreatment with PF-429242. **(E)** Quantification of the differential protein expression, normalized to β-actin (*, $P < 0.5$; **, $P < 0.1$; ***, $P < 0.0001$). **(F)** Normalized Fluo-4 intensity of individual (gray) and averaged (red/blue) time courses from NPC1^I1061T fibroblasts (red, $n = 14$) and NPC1^I1061T fibroblasts (blue, $n = 12$) treated with the same experimental protocol as A. **(G)** Quantification of the AUC during ionomycin. **(H)** Normalized Fluo-4 time series of healthy fibroblasts (black), NPC1^I1061T fibroblasts (red), NPC1^I1061T fibroblasts (red), and NPC1^I1061T fibroblasts treated with PF-429242 for 24 h (blue). **(I)** Quantification of the AUC during TG. **(J)** Quantification of the AUC during SOCE. **(K)** Comparative ratios of the SOCE to TG areas under the curve. Error bars represent the standard error of the mean.

## Inhibition of NPC1 alters lysosomal Ca²⁺ and pH homeostasis via the SREBP pathway

Until now, we have provided evidence that inhibiting NPC1-cholesterol transport from lysosome membranes to ER membranes can misregulate ER Ca²⁺ signaling via the SREBP pathway. Next, we asked if NPC1 dysfunction influences the ability of the lysosome itself to release Ca²⁺ and maintain pH,

two key functions that influence lysosome ability to participate in signaling cascades. To measure lysosomal Ca²⁺ release we overexpressed a genetically encoded Ca²⁺ indicator targeted to lysosomal membranes (GCaMP3-ML1; Fig. 7 A; Shen et al., 2012). Treatment of cells expressing GCaMP3-ML1 with the lysosomal TRPML agonist ML-SA1 (10 µM) caused a rapid release of Ca²⁺ from the lysosome lumen into the cytoplasm (Fig. 7 B). Cells

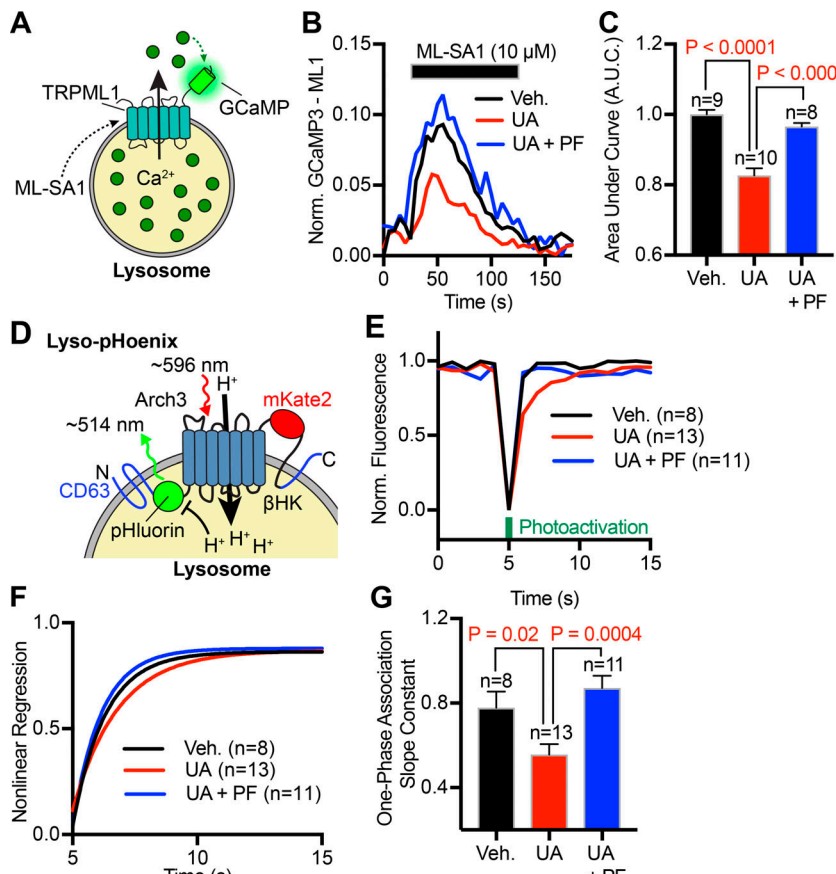

Figure 7. **Inhibition of SREBP rescues lysosomal function following inhibition of NPC1-choelsterol transport. (A)** Schematic of GCaMP3-ML1. **(B)** Representative, normalized changes in GCAMP3-ML1 intensity from tsA201 cells treated with a vehicle control (black, n = 9), UA (red, n = 10), or both UA and PF-429242 (blue, n = 8). **(C)** Quantification of the AUC following addition of ML-SA1. **(D)** Schematic of lyso-pHoenix. **(E)** Representative, normalized changes in lyso-pHoenix intensity from tsA201 cells treated with a vehicle control (black), UA (red), or both UA and PF-429242 (blue). **(F)** One-phase association curves fitted to the average fluorescence recovery of pHoenix following photoactivation. **(G)** Quantitative comparison of the slope constants (K value) from nonlinear regression curves. Error bars represent the standard error of the mean.

incubated overnight with UA (1 µM) to block NPC1-mediated cholesterol efflux had significantly lower Ca²⁺ release (Fig. 7, B and C). This data aligns well with other groups that report reduced lysosomal Ca²⁺ release in NPC1 disease (Lloyd-Evans et al., 2008; Shen et al., 2012). To determine if the ability of lysosomes to regulate pH is also compromised after inhibition of NPC1, we took an optogenetic approach by overexpressing a genetically encoded, light-activated proton pump targeted to lysosomes (lyso-pHoenix; Fig. 7 D; Rost et al., 2015). Upon activation with red light, Arch3 pumps protons into the lysosome lumen to protonate a pH-sensitive GFP (pHluorin), thereby sensitizing its emission. Subsequent removal of red light allows endogenous homeostatic mechanisms (e.g., proton leak) to restore lysosome pH within a few seconds (Fig. 7 E). Treating cells with UA (1 µM) overnight significantly slowed the deprotonation of pHluorin (Fig. 7, E–G), suggesting a reduced proton leak. Critically, cotreatment with a SREBP inhibitor rescued both lysosomal Ca²⁺ release (Fig. 7, B and C) and lysosomal pH homeostasis (Fig. 7, E–G), further underscoring the significant role NPC1-mediated cholesterol–SREBP interactions play in tuning organelle function. Appropriate control experiments detailing the effect of PF-429242 treatment alone are found in Fig. S5, J–M.

## Inhibition of NPC1 recapitulates the Ca²⁺ phenotypes of NPC1^I1061T fibroblasts in primary hippocampal neurons

As noted earlier, patients with a NPC1^I1061T mutation develop NPC1 neurodegenerative disease, raising two questions: is the mechanism described herein conserved in neurons and, if so,

what are the implications of altered neuronal SOCE for NPC disease? To begin addressing these questions, we treated cultured hippocampal neurons with UA to recapitulate the cellular cholesterol phenotype of NPC disease (Fig. 8 A) and measured neuronal Ca²⁺ signals at dendritic spines (Fig. 8 B). Reminiscent of NPC1^I1061T fibroblasts, UA-dependent accumulations in neuronal lysosome cholesterol resulted in a parallel decrease in ER Ca²⁺ levels (Fig. 8 C), increased SOCE (Fig. 8, D and E), and increased resting cytoplasmic Ca²⁺ concentrations (Fig. 8 F).

In neurons, ER Ca²⁺ release and STIM- and ORAI-mediated SOCE reportedly play important roles in the formation and maturation of dendritic spines (Korkotian et al., 2017). Indeed, enhanced SOCE has been shown to mediate dendritic spine loss in an Alzheimer's model of neurodegeneration (Zhang et al., 2015). Thus, we wanted to test if inhibition or disease mutations in NPC1 alter dendritic spine density. To do so, cultured hippocampal neurons (day in vitro [DIV] 21) were fixed and immunostained for the neuronal marker MAP-2 (blue) and dendritic spine marker PSD-95 (green). Fig. 7 G shows representative images of neurons treated for 24, 48, 72, or 96 h with 1 µM UA or vehicle control (time 0). Quantification of images revealed that 24-h UA treatment significantly increased the density of dendritic spines by 48% compared with vehicle-only controls (Fig. 7 G). This initial increase in spine density is likely driven by SOCE, as it could be recapitulated by 24-h TG treatment (yellow panel, Fig. 8 G) and antagonized by blocking ORAI1 channels (purple panel, Fig. 8 G). Prolonged 48- to 96-h incubation with UA led to a progressive, time-dependent decrease in

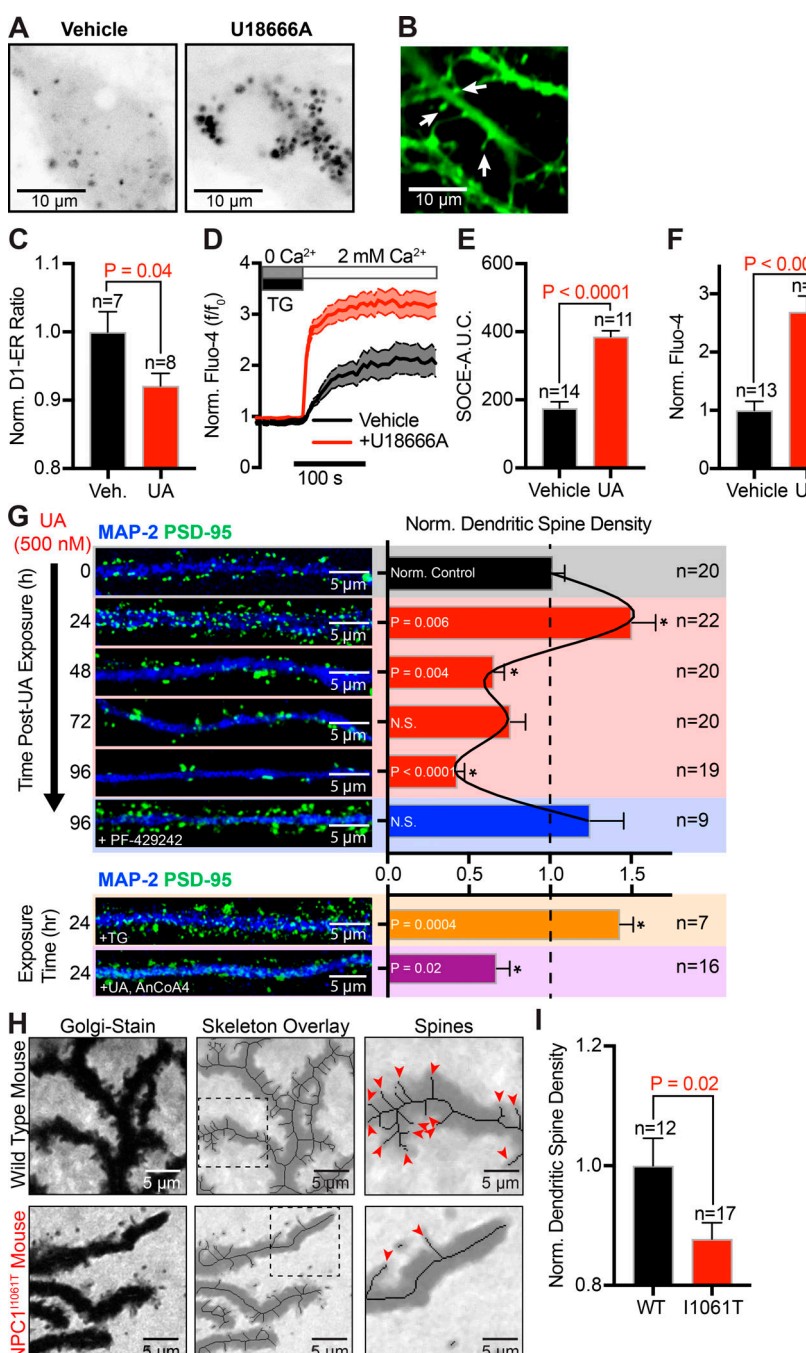

Figure 8. **Inhibition of NPC1 and SREBP pathway recapitulates NPC1[I1061T] disease phenotype in primary neurons and decreases dendritic spine density.** **(A)** Representative superresolution Airyscan images of hippocampal neurons treated with vehicle or NPC1 inhibitor, UA, for 24 h and then fixed and stained with filipin. **(B)** Representative live confocal image of hippocampal neurons loaded with Fluo-4; arrows indicate examples of dendritic spines. **(C)** Quantification of normalized D1-ER FRET ratio from vehicle-treated and UA-treated (24 h) hippocampal neurons at rest. **(D)** Normalized Fluo-4 SOCE traces from dendritic spines in hippocampal neurons vehicle treated (black, $n = 14$) or UA treated for 24 h (red, $n = 11$). **(E)** Quantification of the AUC for the SOCE portion of the time course. **(F)** Comparison of standardized resting Fluo-4 intensity from vehicle-treated and UA-treated (24 h) hippocampal neurons at rest. **(G)** Left: Representative images of hippocampal neurons fixed and immunostained for MAP-2 (blue) and PSD-95 (green) at time points of 0-, 24-, 48-, 72-, and 96-h treatment with UA, 96-h treatment with UA and PF-429242, 24-h treatment with TG, and 24-h treatment with UA and AnCoA4. Right: Quantification of dendritic spine density (number per micrometer), normalized to vehicle control density; asterisk denotes statistical significance. **(H)** Left: Representative images of Golgi-stained Purkinje neurons from WT mice and NPC1[I1061T] mice. Middle: Binary skeleton overlaid each respective representative image. Right: Zoom from dashed box region of interest; arrows indicate examples of spines from the generated skeleton. **(I)** Quantification of dendritic spine density (number per micrometer), normalized to WT density. Error bars represent the standard error of the mean.

spine density (48 h, 66% decrease; 72 h, 60% decrease; 96 h, 69% decrease), which could be rescued by SREBP inhibition (PF-429242; 96 h, 23% increase). We hypothesize that the time-dependent decreases in synaptic spine density between 48-h and 96-h UA treatment is likely an abrupt manifestation of the disease phenotype that likely happens over the course of years in patients. To test this supposition, we fixed and Golgi-stained brain slices from a NPC1[I0611T] murine animal model that faithfully recapitulates patient phenotypes (Praggastis et al., 2015). Analysis of cerebellar Purkinje neurons, the class of cells most susceptible to cell death in NPC1 disease, revealed a significant decrease in spine density between WT and sex-/aged-matched NPC1[I1061T] littermates (WT: $n = 2$, $n = 12$; I1061T: $n = 2$, $n = 17$;

Fig. 8, H and I). These data support a molecular link between NPC1-mediated egress of cholesterol from lysosomes and $Ca^{2+}$-dependent modification of dendritic spine density, which may underlie the disease progression and neurological symptoms observed in NPC disease.

## Discussion

Mechanisms that link cholesterol metabolism to regulation of cellular $Ca^{2+}$ homeostasis remain poorly defined despite their disruptions being strongly correlated with the development and progression of pathophysiological conditions (Liu et al., 2010; Vance, 2012). Here, we show that the lysosomal cholesterol

transporter NPC1 has the ability to alter Ca²⁺ homeostasis and SOCE in patient fibroblasts and primary hippocampal neurons. We have found that the NPC1^I1061T mutation, which reduces cholesterol efflux from lysosomes and results in the most prevalent form of the neurodegenerative NPC1 disease, leads to (1) differential expression of key Ca²⁺ signaling and SOCE regulating proteins, (2) a reduction in luminal ER Ca²⁺, (3) an increase in resting STIM1–ORAI1 interactions, and (4) enhanced SOCE, leading to (5) increased cytoplasmic Ca²⁺ concentrations. Importantly, we also report that inhibition of NPC1 results in a decrease in neuronal spine density. Our experimental evidence suggests that these changes are mediated through the SREBP pathway. This hypothesis is supported by data showing that SREBP pathway inhibition normalizes mRNA and protein expression profiles back to normal ranges and, in parallel, rescues Ca²⁺ signaling in NPC1^I1061T fibroblasts. Collectively, these data suggest that lysosomal NPC1 protein is a key gatekeeper in not only cholesterol metabolism but also Ca²⁺ homeostasis and synaptic architecture.

## SREBP-dependent changes in PSEN1 regulate Ca²⁺ homeostasis

SREBPs are a family of transcription factors often described as master regulators of lipid homeostasis through their ability to acutely sense the concentration of ER cholesterol and maintain its concentration within a precise physiological range (Brown and Goldstein, 1997). SREBPs not only control cholesterol synthesis with remarkable "switch-like" precision (Radhakrishnan et al., 2008) but also are implicated in the direct (via the sterol response element motif) or indirect regulation of ~1,500 protein-coding genes with a variety of functions (Rome et al., 2008). We have provided novel evidence to suggest that Ca²⁺ handling proteins with important roles in Ca²⁺ homeostasis and SOCE can have their protein expressions altered via SREBP. Of note, we find that PSEN1 (discussed below), STIM1, and ORAI1 can have their mRNA and protein abundances increased by SREBP in circumstances of reduced NPC1 cholesterol egress (such as NPC disease) to decrease ER Ca²⁺ concentrations and subsequently increase SOCE.

PSEN1 is an interesting protein at the center of this SREBP-dependent regulation of Ca²⁺ homeostasis in NPC1 disease. Our evidence for the involvement of PSEN1 is robust: (1) PSEN1 is elevated in NPC1^I1061T patient cells, (2) PSEN1 siRNA is sufficient to abrogate the changes in ER luminal Ca²⁺ and SOCE, (3) overexpression of the holoprotein recapitulates NPC1 ER Ca²⁺ loss, and (4) PSEN1^−/− cells do not exhibit UA-dependent changes in Ca²⁺ signaling. PSEN1 has been proposed to act as low-conductance, passive ER Ca²⁺ leak channel, with mutations in the PSEN1 protein leading to disruption or abolishment of ER Ca²⁺ leak activity in familial Alzheimer's disease (Tu et al., 2006; Nelson et al., 2007, 2011). However, such a role for PSEN1 or PSEN2 is controversial (Shilling et al., 2012) and has been challenged by groups who determined that PSEN1 can interact with inositol triphosphate receptors (IP₃R) to prolong IP₃R opening and Ca²⁺ leak permeability (Cheung et al., 2008; Shilling et al., 2014), while others have reported that mutant PSEN1 increases the expression and recruitment of ryanodine receptors to regulate the IP₃R Ca²⁺ signaling in primary

neurons (Stutzmann et al., 2006). Our PSEN1 data fit a simple model in which its increased expression leads to an enhanced intrinsic ER Ca²⁺ leak to decrease ER Ca²⁺ concentrations to a level (~170 µM) that facilitates continuous interactions between STIM1 and ORAI, leading to constitutive Ca²⁺ entry (Luik et al., 2008). Thus, instead of the Ca²⁺ entering the cytoplasm through ORAI1 channels, being pumped via SERCA back into the ER lumen to terminate SOCE, it simply leaks out to perpetuate a feedforward Ca²⁺ signaling loop. Despite this evidence, further experiments are required to fundamentally address the relationship between PSEN1 and other Ca²⁺ release channels in NPC1 disease.

Further underscoring the importance of PSEN1, we demonstrate that knocking it down reduces cholesterol accumulation in NPC1^I1061T disease fibroblasts. This is in line with other groups who demonstrate that targeting ER proteins, including SERCA (Lloyd-Evans et al., 2008), ryanodine receptors (Yu et al., 2012), or calreticulin (Wang et al., 2017), can clear metabolites from cells with lipid storage deficiencies. Mechanistically, elevations in ER Ca²⁺ levels likely remodel the protein-folding environment in the ER to increase steady-state levels of NPC1 (Yu et al., 2012). These data underscore the need for further work detailing the mechanistic underpinnings and timeline of progression of this Ca²⁺ phenotype, as it may be therapeutically beneficial for NPC patients and others with lysosomal storage disorders.

## Consequences of decreased ER Ca²⁺ and enhanced SOCE for cell signaling and neurodegeneration

Given the role of Ca²⁺ as a ubiquitous second messenger, it is clear that deviant elevations in intracellular Ca²⁺ can have detrimental effects on cell health and survival. This is highlighted by the Ca²⁺ hypothesis for neurodegeneration, which postulates that disruption of intracellular Ca²⁺ signaling is pivotal for the molecular mechanisms underlying neuronal degeneration (Biessels and Gispen, 1996; Berridge, 2010; Alzheimer's Association Calcium Hypothesis Workgroup, 2017). Thus, inhibition of lysosomal cholesterol efflux leading to activation of SREBP and downstream changes in local Ca²⁺ gradients could very well be toxic to cells. Underscoring this idea, UA-treated hippocampal neurons and NPC1^I1061T brain slices exhibit reductions in neuron spine density that are time locked with dysfunctional Ca²⁺ signaling. Given that Ca²⁺ is a crucial regulator of dendritic spine plasticity (Higley and Sabatini, 2012), and there is a strong correlation between reduction/elimination of spines and age-dependent memory loss and neurodegeneration (Walsh and Selkoe, 2004; Knobloch and Mansuy, 2008; Marcello et al., 2012; Yu and Lu, 2012; Bezprozvanny and Hiesinger, 2013), it follows that mechanisms controlling spine Ca²⁺ homeostasis may be important for progression of neurodegenerative conditions. In fact, there is growing evidence linking dysfunction of ER Ca²⁺ homeostasis and/or SOCE to alterations in dendritic spine density across several neurodegenerative disorders (Sugawara et al., 2013; Sun et al., 2014; Zhang et al., 2015, 2016; Herms and Dorostkar, 2016; Tong et al., 2016; Wu et al., 2016; Korkotian et al., 2017). Our data provide a testable framework to determine if altered ER Ca²⁺ signaling and synapse loss and/or dendritic regression contributes to cognitive issues in animals with NPC disease.

In conclusion, we report a molecular pathway that links NPC1-mediated efflux of cholesterol from lysosomes to tuning of cytoplasmic $Ca^{2+}$, ER $Ca^{2+}$, and SOCE. Connecting cholesterol efflux to changes in $Ca^{2+}$ is the SREBP pathway, which is active after inhibition or disease mutations in the NPC1 protein, to differentially regulate key $Ca^{2+}$ signaling proteins, including PSEN1. When active, this pathway increases dendritic spine $Ca^{2+}$ concentrations to modulate synaptic plasticity. This works furthers our understanding of lysosome signaling networks and offers a putative contributory mechanism to the neuropathology of NPC disease and potentially other cholesterol-linked neurodegenerative disorders.

## Materials and methods

### Cell culture
Fibroblast cell lines derived from an apparently healthy male (GM05659) and a male patient with the NPC1[I1061T] mutation (GM18453) were purchased from Coriell Institute. Additional fibroblast cell lines derived from patients, NPC1[I1061T, I1061T] and NPC1[I1061T, P1007A], were generously provided by Dr. Forbes D. Porter (National Institutes of Health, Bethesda, MD). Fibroblasts were cultured in MEM supplemented with 2 mM L-glutamine, 15% non–heat-inactivated FBS, and 0.2% penicillin/streptomycin. Cells were passaged twice weekly and incubated in 5% $CO_2$ at 37°C. PSEN[+/+] (WT) and PSEN1/PSEN2 double-knockout (PSEN[−/−]) MEFs were a generous gift from Drs. David Kang, Angels Almenar, and Lawrence S.B. Goldstein (University of California, San Diego, San Diego, CA). WT, NPC1[−/−], and SCAP[−/−] CHO cells were kindly provided by Daniel Ory (Washington University, St. Louis, MO). MEFs, CHO cells, and tsA201 cells were cultured in DMEM supplemented with 10% FBS and 0.2% penicillin/streptomycin. Hippocampal Neurons were isolated from rats at gestation day 18. Neurons were cultured in Neurobasal (21103-049; Gibco) supplemented with B27 (17504-044; Gibco), Glutamax (35050-061; Gibco), 5% FBS, and 0.2% penicillin/streptomycin. On DIV 7, cytosine-D-arabinofuranoside (251010; Millipore) at 1:1,000 was added to neuronal cultures to inhibit astrocyte growth.

### Murine model of NPC1 disease
C57BL/6 WT mice and NPC1[I1061T] knock-in mice (Praggastis et al., 2015) were kindly provided by Daniel Ory. The animal handling protocol was approved by the University of California Institutional Animal Care and Use Committee. Mice were fed standard chow and water ad libitum and housed in a vivarium with controlled conditions. Genotyping of NPC1[I1061T] mice was performed by isolating DNA from mice tails and using PCR with the following primers: NPC1 forward, 5′-TGATCTGCACACTTG GAACCGAG-3′; and NPC1 reverse, 5′-CACTGCCTTGAGCAGCAT CTCAG-3′. Genotype was identified by a 200-bp fragment for WT and a 234-bp fragment for NPC1[I1061T]. Animals used in the study were from symptomatic time points (postnatal day 85–95).

### Filipin staining and AiryScan superresolution imaging
Cells were washed with PBS and fixed in 3% PFA and 0.1% glutaraldehyde for 10 min at 21°C. Cells were then incubated with sodium borohydrate (10 mM) for 5 min and subsequently stained with filipin (3 mg/ml in PBS) for 2 h at room temperature or overnight at 4°C. Cells were excited using a 405-nm LED and light collected using a Plan-Apochromat 63×/1.40 oil-immersion lens and a Zeiss 880 Airyscan microscope. Images were acquired in PBS solution at room temperature (21–23°C) using Zen software.

### $Ca^{2+}$ imaging
Cells were incubated for 30 min in 5 µM Fluo-4 in 2 mM $Ca^{2+}$ Ringer's solution (160 mM NaCl, 2 mM $CaCl_2$, 1 mM $MgCl_2$, 2.5 mM KCl, 10 mM Hepes, and 8 mM glucose) with 0.1% pluronic acid to permeabilize cells. Cells were then moved to a Fluo-4–free Ringer's solution to deesterify for 30 min. Following deesterification, Fluo-4–loaded cells were excited with a 488-nm laser and the resulting fluorescence monitored using an inverted microscope with a Plan-Apochromat 63×/1.40 oil objective, connected to an Andor W1 spinning-disk confocal with a Photometrics Prime 95B camera. Images were acquired every 2–5 s in Ringer's solution at room temperature (21–23°C) using Micromanager software. Hippocampal neurons were imaged at DIV 21; $Ca^{2+}$ measurement regions of interest in neurons were restricted to dendritic spines. SOCE was induced by initially perfusing cells in 2 mM $Ca^{2+}$ solution; cells were then perfused in $Ca^{2+}$-free solution (160 mM NaCl, 2 mM EGTA, 1 mM $MgCl_2$, 2.5 mM KCl, 10 mM Hepes, and 8 mM glucose), followed by 500 nM TG in $Ca^{2+}$-free solution. After 5 min of TG incubation, cells were exposed to 2 mM $Ca^{2+}$ solution to measure resulting SOCE. Resting Fluo-4 intensities were measured by normalizing to the minimum fluorescence in an ionomycin-containing $Ca^{2+}$-free solution. $Ca^{2+}$ dyes and the D1-ER FRET probe were calibrated using standard empirical calculations as previously described (Palmer et al., 2004).

### Transfections, plasmids, and siRNA
The following cDNA plasmids were used in this study (in order of appearance): pGP-CMV-GCaMP6s-CAAX (plasmid 52228; Addgene), mCherry-STIM1 (kind gift from Richard Lewis, Stanford University, Stanford, CA), ORAI1-GFP (Dickson et al., 2012), D1-ER-CAM (Palmer et al., 2004), PSEN1 and PSEN1-D257A (kind gifts from Ilya Bezprozvanny, University of Texas Southwestern, Dallas, TX), GCaMP3-ML1 (kind gift from Haoxing Xu, University of Michigan, Ann Arbor, MI; Shen et al., 2012), and lyso-pHoenix (plasmid 70112; Rost et al., 2015; Addgene). The DsiRNA Duplex targeting PSEN1 was from Integrated DNA Technologies. Transfections of cDNA or DsiRNA were prepared using Lipofectamine LTX (15338-030; Invitrogen) and Lipofectamine RNAiMAX (13778-030; Invitrogen), respectively, in Opti-MEM (31985-062; Gibco). Transfections were incubated overnight for 24 h before imaging, unless otherwise noted.

### Ground state depletion superresolution imaging
Fibroblasts were fixed in 4% PFA for 15 min followed by permeabilization with 0.1% Triton X-100 for 10 min, blocked with 4% FBS for 1 h at 21°C, and stained with rabbit polyclonal anti-ORAI1 (ACC-060, 1:200; Alomone) or rabbit polyclonal anti-STIM1

(HPA012123, 1:1,000; Sigma-Aldrich) overnight at 4°C. Fibroblasts were then washed and incubated for 1 h in Donkey Anti-Rabbit Alexa Fluor 647 (1:1,000; Invitrogen) secondary antibody. Superresolution imaging was performed as previously described (Dixon et al., 2017) using a Leica GSD superresolution microscope. Representative images are displayed at 20-nm pixel size; particle analysis was conducted using 10-nm pixel size.

## Protein extraction and Western blot

Protein from cultured cells was harvested using RIPA Buffer (89900; Thermo Fisher Scientific) with complete, Mini, EDTA-free protease inhibitor (11836170001; Roche) for 25 min at 4°C. Samples were sonicated, and postnuclear supernatant was isolated by centrifugation for 25 min at 13,200 $g$ at 4°C. Concentration of protein lysate was quantified using Pierce BCA protein assay kit (23225; Thermo Fisher Scientific). Protein samples were resolved in 4–12% Bis-Tris gels under reducing conditions. Proteins were transferred from gels onto 0.2-μm polyvinylidene difluoride membranes (#LC2000; Life Technologies) using a Mini-Bolt system (A25977; Thermo Fisher Scientific). using rabbit monoclonal anti-NPC1 (ab134113, 1:2,000; Abcam), rabbit polyclonal anti-ORAI1 (ACC-060, 1:200; Alomone), rabbit polyclonal anti-STIM1 (HPA012123, 1:1,000; Sigma-Aldrich), rabbit polyclonal anti-STIM2 (ACC-064, 1:200; Alomone), mouse monoclonal anti-SERCA2 (ab2861, 1;1,000; Abcam), mouse monoclonal β-actin (MA1-91399 1:10,000; Abcam) Calreticulin (PA3-900; Thermo Fisher Scientific), mouse monoclonal anti-SREBP1 (MA5-11685; Invitrogen), rat monoclonal anti-PSEN1 N-terminal fragment (MAB1563; MilliporeSigma), and mouse monoclonal anti-PSEN1 C-terminal fragment (MAB5232; MilliporeSigma). Blot bands were detected using fluorescent secondary antibodies goat anti-rabbit 680RD (P/N 926-68071, 1:10,000; LI-COR), goat anti-Mouse 800CW (P/N 925-32210, 1:10,000; LI-COR), and goat anti-rat 680 (A-21096, 1:10,000; Invitrogen). ImageJ was used to calculate the fluorescent density of each band. Abundance of proteins were normalized to β-actin; treatment group protein expression was normalized to protein abundance from the appropriate controls.

## mRNA extraction and quantitative real-time PCR

mRNA was isolated from fibroblasts using RNeasy Plus Mini Kit (74134; Qiagen). cDNA was synthesized from 1 μg of total RNA using AffinityScript cDNA Synthesis Kit (600559; Agilent). Quantitative real-time PCR was performed using PowerUp SYBR Green Master Mix (A25741; Applied Biosystems). C-terminal fragment values were normalized to β-actin. The following primer sequences were used: β-actin forward, 5′-AGAGCTACG AGCTGCCTGAC-3′; β-actin reverse, 5′-AGCACTGTGTTGGCG TACAG-3′; STIM1 forward, 5′-AGTCACAGTGAGAAGGCGAC-3′; STIM1 reverse, 5′-CAATTCGCGAAAACTCTGCTG-3′; ORAI1 forward, 5′-GACTGGATCGGCCAGAGTTAC-3′; ORAI1 reverse, 5′-GTCCGGCTGGAGGCTTTAAG-3′; PSEN1 forward, 5′-GAAGCT CAAAGGAGAGTATCCA-3′; PSEN1 reverse, 5′-CCCTAGATGACT GTCCCTCT-3′; PSEN1 forward, 5′-GAAGCGTATACCTAATCT

GGGA-3′; and PSEN1 reverse, 5′-CACAGAAAACAAAGCCTCTTG AG-3′.

## Immunocytochemistry of dendritic spines

Cultured hippocampal neurons were initially washed with PBS and fixed at DIV 21 in 4% PFA for 15 min followed by permeabilization with 0.1% Triton X-100 for 10 min. The neurons were then blocked with 4% FBS for 1 h at 21°C and stained with rabbit polyclonal anti-MAP-2 (AB5622-I, 1:1,000; MilliporeSigma) and mouse monoclonal anti-PSD-95 (75-028, 1:500; Neuromab) overnight at 4°C. Neurons were then washed and incubated 1 h in goat anti-rabbit Alexa Fluor 647 (1:1,000; Invitrogen) and Goat anti-mouse Alexa Fluor 555 (1:1,000; Invitrogen) secondary antibodies and subsequently imaged in PBS using a Plan-Apochromat 63×/1.40 oil DIC M27 objective (superresolution; 880 Airyscan; Zeiss). Images were acquired at room temperature (21–23°C) using Zen software.

## Golgi staining of cerebellar slices

Cerebellums were isolated from paired age- and sex-matched WT and NPC1[I1016IT] mice between 70 and 90 d old. Golgi staining was performed using an FD Rapid GolgiStain Kit (PK401A; FD NeuroTechnologies). Slices were imaged with transmitted light using a 63×/1.40 oil objective connected to an Andor W1 spinning-disk confocal coupled to a Photometrics Prime 95B camera; only Purkinje neurons were assessed for imaging. ImageJ was used to develop representative binary skeletons for quantitative analysis of dendritic spine branching.

## Reagents

UA (Sigma-Aldrich) was dissolved in DMSO and used at a final concentration of 1 μM. AnCoA4 was dissolved in DMSO and used at a final concentration of 50 μM. PF-429242 (Tocris) was dissolved in DMSO and used at a final concentration of 500 nM. TG (Sigma-Aldrich) was dissolved in DMSO and used at a final concentration of 500 nM. Ionomycin (Sigma-Aldrich) was dissolved in DMSO and used at a final concentration of 2.5 to 5 μM. ML-SA1 (Sigma-Aldrich) was dissolved in DMSO and used at a final concentration of 10 μM.

## Data analysis

Microsoft Excel, GraphPad Prism, and IGOR PRO were used to analyze all data. ImageJ was used to process and analyze images. Parametric Student's $t$ tests were conducted to determine significance; P values <0.05 were considered as statistically significant. P values and number of cells are detailed in each figure or figure legend.

## Online supplemental material

Fig. S1 shows impaired cholesterol distribution, ER Ca²⁺, and SOCE across multiple models of NPC1 disease. Fig. S2 shows enhanced $I_{CRAC}$ current and STIM1/ORAI1 activation in NPC1 disease fibroblasts. Fig. S3 demonstrates the involvement of PSEN1 as a regulator of ER Ca²⁺ levels in healthy and NPC1 disease cells. Fig. S4 shows the involvement of the SREBP pathway in mediating transcriptional and functional changes in

$Ca^{2+}$ signaling in NPC1 disease. Fig. S5 shows the effect of SREBP inhibitor PF-429242 in appropriate control cells.

## Acknowledgments

We thank members of the Dickson and Dixon laboratories for the helpful advice and comments. We are extremely grateful to those laboratories (see Materials and methods) that shared reagents, plasmids, and cells lines used in this study. Finally, special thanks to the attendees of the Michael, Marcia, and Christa Parseghian Scientific Conference for NPC Research for their insightful comments and suggestions, which motivated several of the experiments found herein.

This work was supported by a Pharmacology National Institutes of Health T32 training award T32GM099608 (S.A. Tiscione); National Institutes of Health grants R01 HL142282 (D.M. Bers), R01 HL06773 (D.S. Ory), and R01 HL085686 (L.F. Santana); American Heart Association grant 15SDG25560035 (R.E. Dixon); National Institutes of Health grant R01 AG063796 (R.E. Dixon); an Ara Parseghian Medical Research Foundation award (E.J. Dickson); University of California funds (E.J. Dickson); and National Institutes of Health grant R01 GM127513 (E.J. Dickson).

The authors declare no competing financial interests.

Author contributions: S.A. Tiscione and E.J. Dickson conceived, designed, and executed experiments; collected, analyzed, and interpreted data; and wrote and revised the manuscript. K.S. Ginsburg executed experiments and revised the manuscript. R.E. Dixon conceived experiments, interpreted data, and revised the manuscript. D.M. Bers, D.S. Ory, and L.F. Santana interpreted data and revised the manuscript. All authors approved the final version of the manuscript for publication and agree to be accountable for all aspects of the work. All listed authors meet the requirements for authorship, and all those who qualify for authorship are listed.

Submitted: 4 March 2019

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
