## [Reviewer comments · The Journal of Cell Biology]

Disease mutations in Niemann-Pick Type C1 (NPC1) alter ER calcium signaling and neuronal plasticity

Scott Tiscione, Oscar Vivas, Kenneth Ginsburg, Donald Bers, Daniel Ory, Luis Santana, Rose Dixon, and Eamonn Dickson

Corresponding Author(s): Eamonn Dickson, University of California

Review Timeline:

Submission Date:	2019-03-04
Editorial Decision:	2019-04-05
Revision Received:	2019-08-12
Editorial Decision:	2019-08-23
Revision Received:	2019-09-04

Monitoring Editor: William Prinz

Scientific Editor: Melina Casadio

Transaction Report:

DOI: <https://doi.org/10.1083/jcb.201903018>

April 5, 2019

Re: JCB manuscript #201903018

Dr. Eamonn James Dickson
University of California
Physiology and Membrane Biology
Tupper Hall
Davis, California 95616

Dear Dr. Dickson,

Thank you for submitting your manuscript entitled "NPC1 cholesterol efflux regulates PSEN1 to tune store-operated calcium entry and synaptic plasticity". Thank you for your patience with the peer review process. The manuscript has been evaluated by expert reviewers, whose reports are appended below. Unfortunately, after an assessment of the reviewer feedback, our editorial decision is against publication in JCB.

You will see that the reviewers have major concerns about the study - in particular regarding the depth of mechanistic investigations and generality of the findings: they are concerned about the reliance on one mutant line (Rev#1 point #1, and this is a concern we editorially also discussed initially at submission; Rev#2 point #3, Rev#3 point #4). They feel that more insight into how SREBP may affect PS1 and the Ca²⁺ signaling proteins would be needed for publication in JCB (Rev#1 point #2, Rev#2 point #4), ask for the documentation of SREBP activation (Rev#2 point #1), and request that you address the discrepancy with prior work reporting inactivation of SREBP signaling in NPC1 mutant cells (Rev#2 point #2). The referees suggest adding direct measurements of ER Ca²⁺ permeability (Rev#1 #3) and assessing the potential contribution of IP₃ receptors (Rev#1 #4) and PSEN1 endoproteolysis in the ER calcium leak (Rev#2 point #5) as well as whether manipulation of SREBP, PS1, and STIM1-Orai1 restore key lysosomal functions essential for cell survival (Rev#1 #10 -- see also Rev#3's comments about the relevance of the findings to neuronal loss). They suggest additional inhibitor studies to confirm the link between ER cholesterol levels and calcium misregulation (Rev#2 point #2). The reviewers have additional important experimental concerns (#5-6 Rev#1, Rev#3 #3, Rev#3's request to validate the SOCE/CRAC measurements with a more rigorous approach, like patch-clamp) and requests for controls (Rev#1 #7-8-9).

The major limits brought up in review (including the lack of detailed mechanistic insight and the need to use more than one mutant line) are considerable and would require a significant amount of work. We editorially discussed these points and agree with the referees. We find their points valid and most are directly relevant to the core conclusions of the study. Although your manuscript reports an intriguing link between the SREBP pathway and calcium signaling in the context of NPC1 mutation/loss-of-function, we feel that the points raised by the reviewers are more substantial than can be addressed in a typical revision period. If you wish to expedite publication of the current data, it may be best to pursue publication at another journal.

Given interest in the topic, we would be open to resubmission to JCB of a significantly revised and extended manuscript that fully addresses the reviewers' concerns and is subject to further peer-review. Please note that priority and novelty would be reassessed at resubmission. If you would like

to resubmit this work to JCB, please contact the journal office to discuss an appeal of this decision or you may submit an appeal directly through our manuscript submission system. In our view, a resubmission to JCB should focus on addressing all technical and experimental points, especially the requests for more patient mutations/using NPC1 KO cells. This is very important to explore the more general significance of the results. Extending the studies of SREBP signaling along the lines suggested by the revs would additionally be needed. We find the reviewers' comments constructive, and the suggested experiments all seem reasonable to us. The main request from the reviewers that seems beyond the scope of the current study in our view is the request for more details about the role of IP3 receptors (this could be discussed at this stage and explored in other future investigations). But we would encourage you to consider all the other points raised and address them in full, to the best of your ability. If you are interested in resubmitting to JCB, we also encourage you to submit a revision proposal in the form of a point-by-point response to the reviewers' remarks, in which you would delineate how each would be addressed. We would editorially discuss this plan and possibly get referee input. Our goal with this process would be to ensure we are all on the same page about the level of revision needed for resubmission of the journal and ensure you do not embark on time- and resource-consuming experiments that may not be sufficient for a successful resubmission. Please feel free to contact us to discuss the reviewers' comments or if you have any questions.

Regardless of how you choose to proceed, we hope that the comments below will prove constructive as your work progresses. We would be happy to discuss the reviewer comments further once you've had a chance to consider the points raised in this letter. You can contact the journal office with any questions, cellbio@rockefeller.edu or call (212) 327-8588.

Thank you for thinking of JCB as an appropriate place to publish your work.

Sincerely,

William Prinz, PhD
Monitoring Editor, Journal of Cell Biology

Melina Casadio, PhD
Senior Scientific Editor, Journal of Cell Biology

Reviewer #1 (Comments to the Authors (Required)):

This study examined the potential mechanism by which altered NPC1 level and function alter Ca²⁺ signaling associated with cellular toxicity and neurodegeneration. The authors conclude that reduced NPC1 leads to a SREBP-mediated increase in PS1, which leads to a reduced ER Ca²⁺, increased STIM1-Orai1 expression and Ca²⁺ influx and toxicity.

Although the topic is of interest and several experiments are in line of the authors conclusions, several important controls are missing, the studies are incomplete, and the physiological significance/lysosomal function is not explored.

1. The authors used a single cell line with mutant NPC1. The role of NPC1 in Ca²⁺ signaling need to be examined further. At the minimum, the different aspects of the altered Ca²⁺ signal should be measure in additional patient derive cell line. In addition, to exclude the possibility of adaptation in

the immortalized cell line, the authors showed determine whether acute knockout/knockdown of NPC1 has similar effects on PS1, STIM1, Orai1, SERCA and Ca²⁺ signaling.

2. The relationship between SREBP, PS1 and Ca²⁺ signaling proteins was not thoroughly examined. As a TF SREBP may affect transcription and or translation of PS1 and the Ca²⁺ signaling proteins. This was not tested on the molecular level, including gene expression/regulation.

3. The authors claimed increased ER Ca²⁺ leak due to increased PS1 but did not actually measured ER Ca²⁺ permeability in either intact or permeable cells. With the ER Ca²⁺ probe the authors should be able to use similar ER Ca²⁺ loads (for example by incubating control and patient cells at different external Ca²⁺) and measure directly ER Ca²⁺ permeability by following ER Ca²⁺ depletion and the effect of deletion PS1 on the permeability.

4. The potential role of the IP3 receptors, the primary Ca²⁺ release channels, in the presumed Ca²⁺ leak is not even considered, let alone examined.

5. Figure 2: For the TIRF experiments to be of significance, the same cells should be stained for both STIM1 and Orai1 and show co-localization of the puncta, bot in resting and store-depleted cells.

6. Figure 3B-C: The kinetic experiments with expressed proteins when shown as normalizes level are problematic since a) the rate is markedly affected by expression level and b) the overexpression will likely not reproduce the native difference in protein level and the different rates may simply reflects cell health. These experiments should be performed by Co-IP of the native proteins in cells treated with thapsigargin for different times in Ca²⁺-free solutions.

7. Figure 5: These experiments are not complete. The authors should show that knockdown of PS1 restored normal STIM1, Orai1, SERCA and SREBP. In addition, important missing controls are the effect of PS1 knockdown and of the PF drug in wild-type cells is not examined and thus the specificity of the effects are not clear.

8. Figure 6: Again, the control of the effect of the drug in wild-type cells is not shown.

9. Figure 7: a) Drug control is missing; b) panels C/D the Ca²⁺ data and not complete. What is the effects of the drug and SREBP on NPC1 level and as important on basal Ca²⁺ and ER Ca²⁺ content. You cannot claim similar effect without showing these parameters.

10. For the findings to be of significance to cellular functions and toxicity the authors should examine several lysosomal functions to show that manipulation of SREBP, PS1 and STIM1-Orai1 restore key lysosomal functions essential for cell survival.

Reviewer #2 (Comments to the Authors (Required)):

In the paper under review authors try to answer a question: how mutations in NPC1 protein are connected to Ca²⁺ dysregulation.

The paper is innovative and provides insights how mutations in gene involved in cholesterol homeostasis can influence and alter ER Ca²⁺ homeostasis. These findings are not exclusively related to a disease state, but potentially to different metabolic states and dietary conditions. Additional experiments however are needed to support their conclusions

1. Authors claim that SREBP pathways is a critical step connecting defects in cholesterol export to induction of Orai1, STIM1 and SERCA. However, the results of this pathways activation are not provided for any experiments.

It is strongly suggested that Westerns for nuclear SREBP are provided to prove activation of the pathway

2. it has been shown (Abi-Molesh et al., 2009) that in NPC1 null cells SREBP pathway is inhibited, further confirmed by higher ER cholesterol levels in NPC1 null cells. However, according to Tiscione et al., it is opposite. In fact, a common test is LDL challenge, which results in SREBP inhibition in normal cells, but not in NPC1 null.

How authors can explain this discrepancy?

3. Authors study fibroblasts from NPC patients (having a point mutation NPC1 I106T) versus healthy controls. Does NPC1 knockout leads to the same consequences in terms of Ca dysregulation?

4. The authors assume that SREBP pathway causes increase in PSEN1 expression. Is there known SRE motifs in PSEN1 promoter? Please measure PSEN1 mRNA levels by qPCR or Northern blot. Does it go up in NPC1 cells?

5. Only holoprotein form of PSEN1 supports ER Ca leak. Is there a difference in PSEN1 endoproteolysis pattern? PSEN-C and PSEN-N fragments needs to be shown on Western blots. It is possible that increase in ER Ca leak occurs by inhibiting PSEN1 cleavage and increase in holoprotein form. This possibility needs to be tested. The authors can test gamma-secretase inhibitors and also express D257A catalytically dead PSEN1 mutant (that still supports Ca leak activity). Such data help to rule out PSEN1 cleavage relevance

6. Fig 6b. Which difference is significant? Not shown.

Reviewer #3 (Comments to the Authors (Required)):

Niemann-Pick Type C (NPC) is a lysosomal storage disease (LSD) caused by defective export of lysosomal cholesterol, resulting in altered cellular signaling and metabolism. In this paper by Tiscione et al, the authors reported that an upregulation of PSEN1, a proposed ER Ca²⁺ leak channel, in NPC fibroblasts resulted in a decrease in the ER Ca²⁺ level, and an increase of store-operated Ca²⁺ entry (SOCE). In a pharmacologically-induced NPC neuronal model, the authors also reported altered densities of dendritic spines. Overall, it is significant to study the effects of NPC1/2 mutations on neuronal (patho)physiology. However, the authors failed to answer the most important question here, as to whether upregulated SOCE is causal to NPC neuronal loss, or such changes are compensatory and beneficial. Given that most of the reported effects were small, e.g., 20-30% changes in the levels, it is likely that these changes are compensatory. In addition, a previous study cited in the paper failed to see the differences in ER Ca²⁺ release in NPC cells (Lloyd-Evans et al. 2008). Furthermore, given that the basal Ca²⁺ levels were altered in NPC cells, which could compromise SOCE Ca²⁺ measurement in such "stressed" cells, it is important that SOCE/CRAC should be measured using a more rigorous method, e.g., patch-clamp. My specific comments are as follows:

1. The title implies a direct, but not transcriptional regulation of PSEN1 by cholesterol.
2. Based on the proposed model, it is the deficiency of ER cholesterol, manifested by SREBP activation that have caused mis-regulation of ER Ca²⁺ and SOCE. Can PF-429242 reduce NPC neuronal loss? Cyclodextrin is known to reduce lysosomal cholesterol accumulation, without affecting ER cholesterol deficiency. What is the effect of Cyclodextrin treatment on the observed Ca²⁺ mis-regulation in NPC cells?
3. Fura-2 ratios and ER Ca²⁺ should be ideally calibrated to real Ca²⁺ concentrations, so that the differences could be put into perspective.
4. It is not clear why the authors did not use NPC1 KO mice in their neuronal studies.
5. Reduced ER Ca²⁺ levels may compromise neurotransmitter/Gq/PLC signaling in NPC cells. Some discussions on this point might be helpful.

Re: JCB manuscript #201903018

Manuscript title: **NPC1 cholesterol efflux regulates PSEN1 to tune store-operated Ca^{2+} entry and synaptic plasticity**

Dear Drs. Casadio and Prinz,

We would like to thank the editorial team and reviewers for their critical and valuable evaluations of our manuscript. In particular, we would like to thank the editors for providing helpful outlines and for their collegial tone, which has been refreshing. We took these critiques seriously and in response, we designed and executed several new experiments and significantly overhauled and revised the manuscript. Undoubtedly, both editor and reviewer recommendations have strengthened our findings and improved the overall presentation of our work to readers. We are confident, based on the positive comments, clearly defined reviewer/editor requests, and significant number of new experiments, that we have now addressed the concerns of reviewers. We include here a point-by-point response to each of the issues raised by the three expert referees and the editors. We believe that that these additions and edits have markedly improved the manuscript and we hope you now find it acceptable for publication in the Journal of Cell Biology.

General Experiments noted by the editors:

E1. "In our view, a resubmission to JCB should focus on addressing all technical and experimental points, especially the requests for more patient mutations/using NPC1 KO cells"

We agree. To address this comment, we took a three-pronged approach, and performed additional experiments on:

1. Two additional NPC1 patient cell lines (NPC1^{I1061T/I1061T} and NPC1^{I1061T/P1007A}; see Supplemental Fig.1 A-C)

2. NPC1^{-/-} cells (see Supplemental Fig.1 D-F)

3. Cells in which NPC1 function was pharmacologically inhibited with acute (24 h) application of (U18666A; Supplemental Fig.1 G-I)). As illustrated in the new Supplementary Fig. 1 (right) all 3 of these new datasets displayed the same dysregulation of Ca^{2+} homeostasis. These new data fully underscore the important role of the NPC1 cholesterol transporter in tuning cellular Ca^{2+} signals.

E2: “Extending the studies of SREBP signaling along the lines suggested by the reviewers would additionally be needed”

We agree. To answer this critique and to further underscore the involvement of SREBP signaling in mediating the Ca^{2+} responses, we conducted extensive new biochemical analyses as described below:

(1) **Western blot analysis** on healthy and NPC1^{I1061T} patient cell lines determined that the ratio of precursor SREBP (ER localized) and mature nuclear form of SREBP are significantly altered (Figure 6B, C). Put simply, the fractional amount of active SREBP in the nucleus is significantly increased in NPC1^{I1061T}. These data are aligned with our hypothesis that SREBP is responsible for the changes in Ca^{2+} signaling reported herein. These data are also in agreement with published work (Abi-Mosleh et al., 2009; Frolov et al., 2003; Maetzel et al., 2014). We have also determined that the SREBP inhibitor we used in our study (PF-429242), inhibits the accumulation of n-SREBP (Fig. 6B, C) in NPC1 patient cells and normalizes protein levels (Fig. 6D, E) and Ca^{2+} responses (Fig. 6 F-K) back to healthy levels. **These data correlate well with our hypothesis that SREBP mediates the changes in Ca^{2+} signaling and suggests that PF-429424 may be able to rescue this NPC1 phenotype – later we report that this inhibitor also alleviates cholesterol accumulation issues in NPC1 cells suggesting the exciting hypothesis that PF-429424 may present a new therapeutic strategy for NPC1 patients.**

(2) **mRNA measurements** were performed using qPCR to test whether SREBP-mediated transcription correlated with elevations in Ca^{2+} proteins. Our results indicate that differential mRNA levels correlated with the observed changes in protein (Supplemental Figure 5 A, B, next page). PSEN1, STIM1, and ORAI1 mRNA levels were all increased in NPC1 patient fibroblasts; treatment of NPC1 patient fibroblasts with SREBP inhibitor PF-429242 resulted in a decrease in the mRNA of these proteins. This supports that the SREBP pathway is upstream of the transcription of these proteins and targeting this pathway can modulate their expression.

In addition, we performed experiments on **SCAP knock-out cells (SCAP^{-/-})** to underscore the role of SREBP in mediating the Ca^{2+} phenotype in NPC1 disease. SCAP (Sterol regulatory element-binding protein cleavage-activating protein) and a protein of the INSIG family, interact with SREBP promoting its ER-membrane retention. In cholesterol-depleted cells SCAP dissociates from INSIG and escorts SREBP from the ER for proteolytic cleavage to stimulate protein synthesis. In SCAP^{-/-} cells, SREBP is no longer escorted by SCAP and thus does not stimulate protein synthesis (Rawson et al., 1998). Moreover, it has been demonstrated that the

total amount of SREBP is significantly reduced in this cell line (Rawson et al., 1999). We have determined that SCAP^{-/-} cells are refractory to UA treatment and do not present with altered Ca²⁺ dynamics (Fig. S5 C-E) like NPC1 patient cells (compare to Supplemental Fig. 1 above).

Collectively, these three independent lines of investigation all point to SREBP increasing mRNA levels to subsequently elevate the protein abundance of key Ca²⁺ signaling proteins.

E3: “The main request from the reviewers that seems beyond the scope of the current study in our view is the request for more details about the role of IP3 receptors (this could be discussed at this stage and explored in other future investigations)”.

We agree with the editors that these studies are beyond the scope of the current study and more suited to a separate investigation. We have extensively discussed the potential involvement of IP₃ receptors in the discussion.

Comments from the Reviewers

Reviewer #1:

1. The authors used a single cell line with mutant NPC1. The role of NPC1 in Ca²⁺ signaling need to be examined further. At the minimum, the different aspects of the altered Ca²⁺ signal should be measure in additional patient derive cell line. In addition, to exclude the possibility of adaptation in the immortalized cell line, the authors should determine whether acute knockout/knockdown of NPC1 has similar effects on PS1, STIM1, Orai1, SERCA and Ca²⁺ signaling.

As noted in comments to the editors (Page 1, E1) we have conducted additional experiments on two additional NPC1 patient cells lines and a NPC1^{-/-} cell line. Across these three new cell lines, the thapsigargin area under the curve (AUC) was significantly reduced and SOCE was significantly greater compared to healthy controls; these findings are fully consistent with the NPC1^{I1061T} patient fibroblasts. To determine if acute loss of NPC1 function resulted in the same phenotype, mouse embryonic fibroblasts (MEF) cells were treated with U18666A, an inhibitor of NPC1 for 24 hours. UA-mediated inhibition of NPC1 resulted in the same Ca²⁺ phenotype (see new Fig. S1, page 1).

2. The relationship between SREBP, PS1 and Ca²⁺ signaling proteins was not thoroughly examined. SREBP may affect transcription and or translation of PS1 and the Ca²⁺ signaling proteins. This was not tested on the molecular level, including gene expression/regulation.

Thank you for this important comment. As noted above, the Editors also agreed that this point should be addressed. See comments to the editors (page 2, E2) for our response including extensive new data.

3. The authors claimed increased ER Ca²⁺ leak due to increased PS1 but did not actually measure ER Ca²⁺ permeability in either intact or permeable cells. With the ER Ca²⁺ probe the authors should be able to use similar ER Ca²⁺ loads (for example by incubating control and patient cells at different external Ca²⁺) and measure directly ER Ca²⁺ permeability by following ER Ca²⁺ depletion and the effect of deletion PS1 on the permeability.

Good suggestion. We have now included data using the D1-ER Ca²⁺ probe to measure the rate of Ca²⁺ leak after the application of thapsigargin in healthy and NPC1 fibroblasts. Traces were fit to a one-phase decay and the resulting decay constants were compared. The one-phase decay constants and time constants are significantly altered in NPC1 supporting an enhanced ER Ca²⁺ leak. (Fig. 4 J-L right). Additionally, we have performed experiments with overexpressed PSEN1 D257A, which lacks catalytic activity and accumulates as the holoprotein form of PSEN1. As you can see from Fig. S4 D-F (right), thapsigargin mediated Ca²⁺ release is reduced and the rate of ER leak is enhanced. These data, coupled with new data from PSEN1^{-/-} cells (See comment #7 below and Fig. S4) underscore the role of PSEN1 in regulating ER Ca²⁺ dysfunction in NPC disease cells.

4. The potential role of the IP₃ receptors, the primary Ca²⁺ release channels, in the presumed Ca²⁺ leak is not even considered, let alone examined.

As discussed in comment E3 to the editors, while we agree that this is an interesting line of enquiry for a future study, it considered beyond the scope of the current study. We have added a section in discussion detailing the potential involvement of IP₃R's.

5. Fig. 2: For the TIRF experiments to be of significance, the same cells should be stained for both STIM1 and Orai1 and show co-localization of the puncta, both in resting and store-depleted cells.

Thank you for this comment. Additional immunofluorescence experiments have been performed on NPC1 patient fibroblasts and healthy controls stained for STIM1 and Orai1. The degree of colocalization was determined using super-resolution confocal microscopy with NPC1 patient fibroblasts having significantly greater density and area of overlap between STIM1 and Orai1 at

rest (Fig. 3 A, B, C). This new figure better supports our hypothesis that resting STIM1-Orai1 interactions are enhanced in NPC1 disease.

6. Figure 3B-C: The kinetic experiments with expressed proteins when shown as normalized level are problematic since a) the rate is markedly affected by expression level and b) the overexpression will likely not reproduce the native difference in protein level and the different rates may simply reflect cell health. These experiments should be performed by Co-IP of the native proteins in cells treated with thapsigargin for different times in Ca^{2+} -free solutions.

We agree with the reviewer. Our initial thought was to remove the data but we ultimately decided to move it to the supplement (Fig. S3) as we believe readers would expect to see this type of overexpression experiment. Given the new Fig. 3 detailing enhanced proximity of endogenous STIM1-Orai (discussed in point 5 above), coupled with new patch-clamp electrophysiology (Fig. S2, discussed in reviewer 3 comments), demonstrating enhanced I_{CRAC} , we believe this data accurately reflects increased STIM1-Orai interactions in NPC1 disease.

7. Figure 5: These experiments are not complete. The authors should show that knockdown of PS1 restored normal STIM1, Orai1, SERCA and SREBP. In addition, important missing controls are the effect of PS1 knockdown and of the PF drug in wild-type cells is not examined and thus the specificity of the effects are not clear.

Experimental controls demonstrating effect of PF on healthy fibroblast has now been included (Fig. S6). Endpoints measured mirror those presented in Fig. 5 including ER Ca^{2+} load and SOCE data with appropriate quantitative analysis. Further, we have now included datasets from healthy fibroblasts treated with siRNA against PSEN1, NPC1^{I1016T} fibroblasts treated with siRNA against PSEN1, and PSEN1^{-/-} cells. Collectively, these data demonstrate that:

(1) PSEN1 is increased in NPC1 disease (Fig. 5A-C).

- (2) Inhibition of NPC1 increases the holoprotein form of PSEN1 (Fig. 5C)
- (3) Treating healthy fibroblasts with siRNA against PSEN1 increases ER calcium (Fig. S4 A-C).
- (4) Treating NPC1^{L1016T} fibroblasts with siRNA directly against PSEN1 normalizes calcium signals (Fig. 5F-H).
- (5) Overexpressing the holoprotein form of PSEN1 (supports Ca²⁺ leak activity) decreases ER calcium and increases rate of ER Ca²⁺ leak (Fig. S4 D-F)
- (6) PSEN1^{-/-} cells have increased ER Ca²⁺ and SOCE (Fig. S4 G-I)
- (7) Treating PSEN1^{-/-} with UA, PF, or UA and PF does not alter Ca²⁺ dynamics (Fig. S4 J-L).

Taken together, these data underscore the importance of the PSEN1 protein in regulating ER Ca²⁺ levels and that its NPC1-dependent upregulation in NPC1 disease is a major determinant of altered Ca²⁺ signaling.

8. Figure 6: Again, the control of the effect of the drug in wild-type cells is not shown.

Thank you for raising this point. Fig. S6 now contains control experiments using healthy fibroblasts treated with SREBP inhibitor PF-429242 or vehicle control. Endpoints measured mirror those presented in Fig. 5 including Western Blot Ca²⁺ protein expressional changes, ER Ca²⁺ load and SOCE data with appropriate quantitative analysis.

9. Figure 7: a) Drug control is missing; b) panels C/D the Ca²⁺ data and not complete. What is the effects of the drug and SREBP on NPC1 level and as important on basal Ca²⁺ and ER Ca²⁺ content. You cannot claim similar effect without showing these parameters.

Thank you for this comment. In order to show the U18666A faithfully recapitulates the NPC1 disease Ca²⁺ phenotype, resting Ca²⁺ was measured using Fluo-4 and ER Ca²⁺ using D1-ER. Treatment with UA resulted in a significant decrease in ER Ca²⁺ (Fig. 8C) increase in SOCE (Fig. 8D), and increase in resting cytosolic Ca²⁺ (Fig. 8F). These data are consistent with NPC1 disease mutations and NPC1^{-/-}

experiments and support the use of this compound as a way to pharmacologically mimic NPC1 disease.

10. For the findings to be of significance to cellular functions and toxicity the authors should examine several lysosomal functions to show that manipulation of SREBP, PS1 and STIM1-Orai1 restore key lysosomal functions essential for cell survival.

Lysosomal function was assessed in three ways:

Lysosomal Ca^{2+} : To measure lysosomal Ca^{2+} release, cells were transfected with a genetically encoded calcium sensor targeted to the external lysosomal membrane (Fig. 7 A, B, C). Cells treated with the NPC1 inhibitor U18666A had significantly decreased lysosomal Ca^{2+} release. This reduction in lysosomal Ca^{2+} was rescued by inhibition of the SREBP pathway with PF-429242. Controls for these experiments appear in Fig. S7.

Lysosomal deacidification. Lysosomes require tight regulation of the luminal pH in order to maintain their function, thus we tested the lysosomal capacity to restore pH by transfecting cells with a light-activated proton pump which acidifies the lysosome. This plasmid contains a pHluorin within the lumen of the lysosome to measure pH (Fig. 7D). Cells were either treated with a vehicle control, U18666A, PF-429242, or both. Treatment with U18666A resulted in a significantly slower time constant for the lysosomal lumen to deacidify following acute activation of the genetically encoded proton pump. PF-429242 co-treatment with UA rescued this prolonged recovery back to control levels (Fig. 7 D, E, F, G). These data demonstrate that targeting SREBP can restore key lysosomal functions in NPC disease. Controls for these experiments appear in Fig. S7.

Lysosomal Cholesterol: As a final measure of lysosome function in NPC1 disease we tested the hypothesis that treating NPC1^{1061T} fibroblasts with PSEN1 siRNA, to increase ER Ca^{2+} levels (Fig. 5 D-G), may increase the amount of NPC1 protein and alter cholesterol storage. Western blot analysis revealed that 24 hrs PSEN1 siRNA treatment of NPC1^{1061T} cells increased NPC1 protein about ~3-fold (Fig. 5 I, J, right) and caused a corresponding decrease in the area and intensity of Filipin-positive vesicles (Fig. 5 K – N). These results demonstrate that increases in PSEN1, which decrease ER Ca^{2+} levels, may compound cholesterol accumulation in NPC1 disease.

Reviewer #2

We thank Reviewer 2 for their helpful comments.

1. Authors claim that SREBP pathways is a critical step connecting defects in cholesterol export to induction of Orai1, STIM1 and SERCA. However, the results of this pathways activation are not provided for any experiments.

Thank you for this comment. Our response to this important suggestion can be found in comments to the editor (E2, page2).

2. it has been shown (Abi-Molesh et al., 2009) that in NPC1 null cells SREBP pathway is inhibited, further confirmed by higher ER cholesterol levels in NPC1 null cells. However, according to Tiscione et al., it is opposite. In fact, a common test is LDL challenge, which results in SREBP inhibition in normal cells, but not in NPC1 null. How authors can explain this discrepancy?

Thanks for this comment. To clarify, the work of Abi-mosleh (Abi-Mosleh et al., 2009) determined that NPC1 patient fibroblasts have active N-SREBP, similar to what we have determined in Fig. 2. As noted above and in comments to the editor (E2, Page 2), we have now measured and determined that NPC1^{I1061T} patients have elevated N-SREBP in resting cells.

3. Authors study fibroblasts from NPC patients (having a point mutation NPC1 I106T) versus healthy controls. Does NPC1 knockout leads to the same consequences in terms of Ca dysregulation?

Excellent suggestion. Yes, NPC1^{-/-} leads to the same calcium dysregulation (Fig. S1 D-F). We have included these results and data from 2 additional NPC1 disease cell lines to underscore the relationship between NPC1 and Ca²⁺ dysregulation

4. The authors assume that SREBP pathway causes increase in PSEN1 expression. Is there known SRE motifs in PSEN1 promoter? Please measure PSEN1 mRNA levels by qPCR or Northern blot. Does it go up in NPC1 cells?

To our knowledge, there are no reporter SRE motifs in the PSEN1 Promoter. As requested by the reviewer qPCR was performed for Healthy and NPC1 patient fibroblasts with either vehicle treatment or SREBP inhibitor (PF-429242) (Supplemental Fig. S5 A,B). These results revealed a significant increase in PSEN1 mRNA in NPC1 patient fibroblasts; coupled with an increase in protein expression, this suggests PSEN1 transcription is increased in NPC1 disease. This trend was also seen in STIM1 and ORAI1 mRNA, which potentiate SOCE. Treatment with SREBP inhibitor PF-429242 results in a reduction of mRNA for PSEN1, STIM1, and ORAI1 in NPC1 fibroblasts. This supports that SREBP is upstream of these transcriptional changes in NPC1 disease. We have added these data and a note in the discussion about SRE motifs.

5. Only holoprotein form of PSEN1 supports Er Ca leak. Is there a difference in PSEN1 endoproteolysis pattern? PSEN-C and PSEN-N fragments needs to be shown on Western blots. It is possible that increase in ER Ca leak occurs by inhibiting PSEN1 cleavage and increase in holoprotein form. This possibility needs to be tested. The authors can test gamma-secretase inhibitors and also express D257A catalytically dead PSEN1 mutant (that still supports Ca leak activity). Such data help to rule out PSEN1 cleavage relevance

Thank you for this excellent suggestion. Please see comments to the editors (E2, page2). Fig. 6 A, B shows the relative expression of the PSEN1-C and PSEN1-N fragments in NPC1 fibroblasts compared to healthy cells, alongside PSEN1siRNA treated fibroblasts as controls. NPC1 fibroblasts have a significant increase in fragment expression; however, the holoprotein form of PSEN1 was not visible in either healthy or NPC1 fibroblasts. This is perhaps not unexpected given its reported low abundance and difficulty in direct detection. To visualize differences in holoprotein, tsA cells were transfected with full length PSEN1 and treated with NPC1 inhibitor or vehicle control. Western blot was performed for PSEN1 and the holoprotein band was visible (Fig. 5 C); inhibition of NPC1 resulted in a significant increase in PSEN1 holoprotein, suggesting the loss of NPC1 function results in both an increase in PSEN1 holoprotein and its terminal fragments. tsA cells were also transfected with catalytically dead PSEN1- D257A and ER Ca²⁺ was measured as the increase in cytosolic Ca²⁺ area under the curve signal following treatment with thapsigargin. Overexpression of PSEN1-D257A resulted in a reduction of ER Ca²⁺, supporting that increases in PSEN1 expression support altered ER Ca²⁺ load (see Fig. S4 D-F). Additionally, these thapsigargin curves were fitted with a Boltzmann sigmoidal nonlinear regression curve. The PSEN1-D257A transfected cells had a lower slope constant, suggesting a larger ER Ca²⁺ leak. Finally, as noted to the Editors and Reviewer 1, PSEN1^{-/-} cells did not exhibit calcium dysregulation when NPC1 was blocked with the UA compound (Fig. S4 J-L). These data represent multiple lines of independent evidence for the involvement of PSEN1 in mediating Ca²⁺ dysfunction in NPC1 disease.

6. Fig 6b. Which difference is significant? Not shown.

Thank you. We have edited our figure for clarity.

Reviewer #3:

We thank the reviewer for their careful reading and insightful comments. Below we address each comment individually. In addition to these comments we now include patch-clamp electrophysiology demonstrating that an ORAI1-sensitive, SOCE conductance (I_{CRAC}) is significantly elevated in NPC1 fibroblasts (Fig.S2, below). Further, as noted to the editors we add data from 2 additional NPC1 patient patients, NPC1^{-/-} cells, and neurons treated with an inhibitor of UA. Importantly, all demonstrate that ER Ca²⁺ is significantly reduced and SOCE elevated. Underscoring that UA treatment recapitulates effects overserved in disease we have obtained data from an NPC1^{I1061T} knock-in mouse model, that faithfully recapitulates the human disease, and determined that spine density is significantly reduced in NPC1 disease. We believe that these additions, along with the numerous other experiments listed above/below underscore a mechanistic link between cholesterol efflux from the lysosome and the tuning of ER calcium levels in health and disease.

1. The title implies a direct, but not transcriptional regulation of PSEN1 by cholesterol.

Thank you for this comment. In the abstract, results, and discussion we emphasize the transcriptional regulation of PSEN1 by cholesterol. We therefore respectfully maintain that the title remain as worded.

2. Based on the proposed model, it is the deficiency of ER cholesterol, manifested by

SREBP activation that have caused mis-regulation of ER Ca²⁺ and SOCE.

In our model, yes, reduced cholesterol efflux from lysosomes to ER membranes results in activation of SREBP which influences protein levels of crucial calcium handling proteins. We now provide additional support for this model (see comments to editors E2).

3. Can PF-429242 reduce NPC neuronal loss?

This is an excellent question and hopefully one that we can address in the future. In an ideal world one would treat NPC1^{I1061T} mice with a SREBP inhibitor and at different time points measure cerebellar neuron numbers. This type of experiment is not trivial and requires large scale, carefully controlled animal tests. We do show that treatment with a SREBP inhibitor (PF-429242) rescues spine density in neurons treated with a NPC1 inhibitor (Fig. 8F). Given that spine density is also reduced in NPC1^{I1061T} neurons (Fig. 8G), it follows that inhibition of SREBP may be a hopeful target to rescue spine density and neuron loss in NPC1 disease. As mentioned, we hope to use the current model to test this hypothesis in the future but believe it is beyond the reasonable scope of experiments for a resubmission. We have incorporated this hypothesis in the discussion section.

4. Cyclodextrin is known to reduce lysosomal cholesterol accumulation, without affecting ER cholesterol deficiency. What is the effect of Cyclodextrin treatment on the observed Ca²⁺ mis-regulation in NPC cells?

This is a complicated question. While there is evidence that cyclodextrin can extract cholesterol from lysosomes, independently of the ER. Other studies have shown that cholesterol released from LEs/Ls of NPC-deficient cells by cyclodextrins reaches the cytosolic compartment and is accessible to the ER. Underscoring the increased cholesterol availability at the ER following cyclodextrin treatment is decreased expression/processing of SREBP (Peake and Vance, 2012). Given the complicated nature of Cyclodextrin treatment we believe this is a topic for future studies.

5. Fura-2 ratios and ER Ca²⁺ should be ideally calibrated to real Ca²⁺ concentrations, so that the differences could be put into perspective.

Thank you for this excellent suggestion. For cytoplasmic Ca²⁺, we have calculated resting values of around 250 nM for NPC1^{I1061T} fibroblasts, approximately 100 nM elevated compared to healthy patient fibroblasts. See text for precise values.

For ER calcium, NPC1^{I1061T} fibroblasts have a significantly lower luminal ER free Ca²⁺ concentration of ~160 μM, compared to healthy fibroblasts (270 μM). These empirical calculations fit nicely with our overall model, that decreased ER Ca²⁺ in NPC1 fibroblasts results in increased

I_{CRAC} (Fig. S2) and perfectly agree with the $[Ca^{2+}]_{ER}$ -response relation for the CRAC channel calculated by Luik et al., (Luik et al., 2008).

4. *It is not clear why the authors did not use NPC1 KO mice in their neuronal studies.*

As mentioned above we have now included data using the NPC1^{I1061T} animal.

5. *Reduced ER Ca²⁺ levels may compromise neurotransmitter/Gq/PLC signaling in NPC cells. Some discussions on this point might be helpful.*

Excellent suggestion. As noted to the editors and each reviewer we have now included a section in the discussion how reduced ER Ca²⁺ levels may compromise neurotransmitter signaling in NPC1 cells.

References

- Abi-Mosleh, L., Infante, R.E., Radhakrishnan, A., Goldstein, J.L., and Brown, M.S. (2009). Cyclodextrin overcomes deficient lysosome-to-endoplasmic reticulum transport of cholesterol in Niemann-Pick type C cells. *Proc Natl Acad Sci U S A* 106, 19316-19321. 10.1073/pnas.0910916106
- Lloyd-Evans, E., Morgan, A.J., He, X., Smith, D.A., Elliot-Smith, E., Sillence, D.J., Churchill, G.C., Schuchman, E.H., Galione, A., and Platt, F.M. (2008). Niemann-Pick disease type C1 is a sphingosine storage disease that causes deregulation of lysosomal calcium. *Nat Med* 14, 1247-1255. 10.1038/nm.1876
- Luik, R.M., Wang, B., Prakriya, M., Wu, M.M., and Lewis, R.S. (2008). Oligomerization of STIM1 couples ER calcium depletion to CRAC channel activation. *Nature* 454, 538-542. 10.1038/nature07065
- Peake, K.B., and Vance, J.E. (2012). Normalization of cholesterol homeostasis by 2-hydroxypropyl-beta-cyclodextrin in neurons and glia from Niemann-Pick C1 (NPC1)-deficient mice. *J Biol Chem* 287, 9290-9298. 10.1074/jbc.M111.326405
- Rawson, R.B., DeBose-Boyd, R., Goldstein, J.L., and Brown, M.S. (1999). Failure to cleave sterol regulatory element-binding proteins (SREBPs) causes cholesterol auxotrophy in Chinese hamster ovary cells with genetic absence of SREBP cleavage-activating protein. *J Biol Chem* 274, 28549-28556. 10.1074/jbc.274.40.28549

August 23, 2019

RE: JCB Manuscript #201903018R-A

Dr. Eamonn James Dickson
University of California
Physiology and Membrane Biology
Tupper Hall
Davis, California 95616

Dear Dr. Dickson,

Thank you for submitting your revised manuscript entitled "NPC1 cholesterol efflux regulates PSEN1 to tune store-operated Ca²⁺ entry and synaptic plasticity". You will see that the reviewers all found that the revisions adequately strengthened the conclusions and addressed their comments. We continue to feel that examining the role of IP3 receptors in future work will be exciting and interesting and that, consistent with our prior decision, these lines of investigation are not required for publication in JCB. We would be happy to publish your paper in JCB pending final revisions necessary to meet our formatting guidelines (see details below).

- 1) Text limits: Character count for Articles and Tools is < 40,000, not including spaces. Count includes title page, abstract, introduction, results, discussion, acknowledgments, and figure legends. Count does not include materials and methods, references, tables, or supplemental legends.
- 2) JCB Articles are limited to 10 main and 5 supplementary figures. Each figure can span up to one entire page as long as all panels fit on the page. Could you please rearrange the data so as to meet this limit? Some of the supplementary data could be combined into fewer figures or moved to the main figures. Please let us know if you would like input on the changes needed at this stage.
- 3) Titles, eTOC: Please consider the following revision suggestions aimed at increasing the accessibility of the work for a broad audience and non-experts.

Title: Disease-associated mutations in Niemann-Pick Type C1 (NPC1) alter ER calcium signaling and neuronal plasticity

eTOC summary: A 40-word summary that describes the context and significance of the findings for a general readership should be included on the title page. The statement should be written in the present tense and refer to the work in the third person.

- Please include an eTOC statement on the title page at resubmission. It should start with "First author et al." to meet our style requirements.

- 4) Figure formatting: Scale bars must be present on all microscopy images, including inset magnifications. Please add scale bars to 1B (magnifications), 2FH (mags), 3AB (mags), 8H (mags), S1 (top images), S3D (mags)

5) Statistical analysis: Error bars on graphic representations of numerical data must be clearly described in the figure legend. The number of independent data points (n) represented in a graph must be indicated in the legend. Statistical methods should be explained in full in the materials and methods. For figures presenting pooled data the statistical measure should be defined in the figure legends.

Please indicate n/sample size/how many experiments the data are representative of: 1C, 4EFJ, 5F, 6FH, 7BEF, 8D, S1ADG, S2ADE, S3BC, S4ADFGJM, S5AC, S6G, S7AC

6) Materials and methods: Should be comprehensive and not simply reference a previous publication for details on how an experiment was performed. Please provide full descriptions in the text for readers who may not have access to referenced manuscripts.

- Please briefly describe the basic genetic features for all cell lines, mouse lines, and constructs, even if given to you by other investigators or described in other published work.
- Please include sequences for all siRNAs, including negative controls, if they were made available to you from the manufacturer.
- Microscope image acquisition: The following information must be provided about the acquisition and processing of images:
 - a. Make and model of microscope
 - b. Type, magnification, and numerical aperture of the objective lenses
 - c. Temperature
 - d. imaging medium
 - e. Fluorochromes
 - f. Camera make and model
 - g. Acquisition software
 - h. Any software used for image processing subsequent to data acquisition. Please include details and types of operations involved (e.g., type of deconvolution, 3D reconstitutions, surface or volume rendering, gamma adjustments, etc.).

7) A summary paragraph of all supplemental material should appear at the end of the Materials and methods section.

- Please include ~1 brief descriptive sentence per item.

A. MANUSCRIPT ORGANIZATION AND FORMATTING:

Full guidelines are available on our Instructions for Authors page, <http://jcb.rupress.org/submission-guidelines#revised>. **Submission of a paper that does not conform to JCB guidelines will delay the acceptance of your manuscript.**

B. FINAL FILES:

-- High-resolution figure and video files: See our detailed guidelines for preparing your production-ready images, <http://jcb.rupress.org/fig-vid-guidelines>.

Thank you for this interesting contribution, we look forward to publishing your paper in the Journal of Cell Biology.

Sincerely,

William Prinz, PhD
Monitoring Editor, Journal of Cell Biology

Melina Casadio, PhD
Senior Scientific Editor, Journal of Cell Biology

Reviewer #1 (Comments to the Authors (Required)):

The authors thoroughly addressed all my comments and concerns, except examining the role of IP3Rs in the enhanced ER Ca²⁺ leak. In my opinion this is quite significant point because IP3Rs are by far the most prominent Ca²⁺ efflux/release proteins in the ER. Not examining their state in the NPC1 cells in view of the increased ER Ca²⁺ leak does not make sense. However, at this stage this is an editorial decision. Short of this issue, the studies are thorough and the manuscript presents novel data that significantly advances understanding the disease.

Reviewer #2 (Comments to the Authors (Required)):

The authors made serious effort to address most of the comments raised by the referees and the paper has been significantly improved as a result. It is an interesting and innovative study that provide another potential links between cholesterol and calcium signaling - 2 key pathways related to neurodegeneration. I have no further comments

Reviewer #3 (Comments to the Authors (Required)):

The authors have done an excellent job in addressing my concerns.